# BESplit: Bias-Compensated Split Federated Learning with Evidential Aggregation

**Yuhan Xie** [1 2]  **Chen Lyu** [1 2]  **Jingrong Huang** [1]

## Abstract

Split Federated Learning (SFL) enables privacy-preserving collaborative training by partitioning models between clients and a server. However, under non-IID data distributions, SFL often suffers from biased optimization and unstable convergence, while existing solutions largely adapt techniques from conventional federated learning. In this work, we observe that the split architecture of SFL inherently alters how client information is represented and coordinated, opening opportunities for bias compensation beyond parameter-level aggregation. Based on this insight, we propose **BESplit**, an architecture-aware framework that exploits the intrinsic structure of SFL to mitigate non-IID effects. First, to prevent biased local data from dominating global updates, we introduce Evidential Aggregation (EA) to perform fine-grained reweighting of client contributions based on evidential uncertainty. Second, to further reduce distributional skew, we develop Bias-Compensated Collaboration (BCC) to align split-layer representations by pairing complementary clients. Finally, Dual-Teacher Distillation (DTD) is incorporated to synchronize knowledge between decoupled client and server models, enabling independent local inference. Extensive experiments on five benchmark datasets demonstrate that BESplit consistently outperforms state-of-the-art methods in accuracy, convergence stability, and computational efficiency under diverse non-IID settings.

---

[1]Shanghai University of Finance and Economics, China [2]MoE Key Laboratory of Interdisciplinary Research of Computation and Economics, China. Correspondence to: Chen Lyu <lyu.chen@mail.shufe.edu.cn>.

*Proceedings of the 43rd International Conference on Machine Learning*, Seoul, South Korea. PMLR 306, 2026. Copyright 2026 by the author(s).

## 1. Introduction

Split Federated Learning (SFL) (Lin et al., 2025; Xie & Lyu, 2026) has recently emerged as an effective collaborative framework that unifies the advantages of Federated Learning (FL) (Wang et al., 2025; Haripriya et al., 2025) and Split Learning (SL) (Pham et al., 2025). By partitioning a neural network into client-side and server-side models, SFL enables distributed training. Specifically, each client executes partial forward propagation and transmits the resulting smashed data (i.e., intermediate representations) to the server. The server then completes the remaining computations and returns gradients for local updates. Both client-side and server-side parameters are subsequently aggregated through distinct coordination servers. This architecture alleviates client-side burdens while preserving data privacy by avoiding raw data exposure (Wu et al., 2024).

Despite its benefits, the decentralized nature of SFL inherently renders it vulnerable to both system and data heterogeneity (Thapa et al., 2022; Tirana et al., 2025). Existing studies (Liao et al., 2024b;a; He et al., 2025; Lin et al., 2026) have predominantly focused on alleviating system heterogeneity, which stems from disparities in clients' computational and communication capabilities, through cooperative or resource-aware training strategies. However, the challenge of data heterogeneity under non-Independent and Identically Distributed (non-IID) settings across clients has received comparatively less attention in SFL. Such non-IID distributions in SFL introduce gradient bias and inconsistent client optimization, resulting in slower convergence, higher computational costs, and degraded global performance (Xiao et al., 2024). Hence, mitigating these non-IID effects is pivotal for achieving practical and scalable SFL.

To address the impact of non-IID data in conventional FL, many prior works have adopted optimization-based strategies such as FedProx (Li et al., 2020), SCAFFOLD (Karimireddy et al., 2020), and FedDyn (Acar et al., 2021), which constrain local updates or adjust gradient directions to align with the global objective. Other studies have explored representation-based approaches, including FedRDN (Yan et al., 2025), FAVD (Kumar et al., 2025), and Fed-MoE (Jiang et al., 2025), to alleviate distributional skew by performing feature augmentation or refining knowledge trans-

fer. Although these strategies developed for conventional FL can, in principle, be adapted to SFL due to their architectural similarities (Xie & Lyu, 2026), their effectiveness within the SFL paradigm remains largely unexplored. We hypothesize that the split architecture of SFL provides inherent advantages for mitigating non-IID effects, which have not yet been systematically exploited. This leads to a fundamental question: *How can this split design be leveraged to handle client-side data heterogeneity more effectively?*

To answer this question, we make two key observations in SFL, each associated with a distinct challenge. First, within the split architecture, the server inherently conducts server-side training based on the smashed data transmitted from participating clients in each communication round. Although privacy-preserving, these smashed data still retain rich statistical information about client distributions (Liao et al., 2024a;b). The key challenge, however, lies in how to effectively learn such representations to capture fine-grained client heterogeneity and improve global aggregation under non-IID conditions. Second, since the server already receives smashed data from all participating clients, these intermediate representations naturally serve as a built-in medium for server-side coordination. By strategically aligning complementary features, the server can effectively compensate for biased local data distributions without additional client-side interaction. While this setup enables potential benefits, the main challenge is how to identify and exploit such distributional complementarity to alleviate non-IID data bias. By leveraging these two characteristics, SFL provides a promising foundation for developing *bias-compensated mechanisms* that move beyond conventional parameter aggregation.

Motivated by these insights, we propose **BESplit**, an architecture-aware SFL framework that systematically addresses non-IID challenges across three complementary dimensions: *aggregation*, *feature*, and *model* levels. An overview of the BESplit components is provided in Table 1 and Fig. 1. First, at the aggregation level, Evidential Aggregation (EA) is designed to stabilize global optimization by explicitly modeling client heterogeneity. For each client, EA introduces a Client State Record (CSR) to collect evidential statistics from smashed data using Evidential Deep Learning (EDL) (Chen et al., 2024b), including both evidence and aleatoric–epistemic uncertainties (Xie et al., 2023; Shen et al., 2023). Based on these CSRs, EA adaptively reweights client updates to suppress the influence of biased local data. Second, at the feature level, Bias-Compensated Collaboration (BCC) is developed to reduce distributional skew by leveraging cross-client complementarity. Specifically, by utilizing smashed data and CSRs available at the server, BCC identifies clients with biased local distributions based on their divergence from the global distribution (Yan et al., 2025). It then pairs these clients according to distributional

*Table 1.* Overview of three components in BESplit. Each module operates at a distinct operation level of SFL and tackles complementary aspects of heterogeneous non-IID data. $\nu$ denotes the average proportion of each module's time cost relative to the total SFL runtime across all datasets.

| Module | Operation Level | Location | $\nu$ |
|---|---|---|---|
| **EA** | Aggregation | Server | 0.01% |
| **BCC** | Feature | Server | 0.12% |
| **DTD** | Model | Client | 3.65% |

complementarity to correct representation bias. Finally, informed by EA and BCC, Dual-Teacher Distillation (DTD) is incorporated to alleviate model divergence caused by decoupled client and server training. Each client learns a lightweight auxiliary model via knowledge distillation from two teachers (Yu et al., 2024b): the local model for feature alignment and the global model for evidential guidance. This design could preserve local specialization while ensuring global consistency, thereby enabling local inference without additional communication for model updates. The main contributions of this work are summarized as follows:

- We explore the rarely studied problem of SFL under data heterogeneity, revealing its distinct challenges beyond conventional FL and offering insights into how to understand and solve this problem.

- We propose BESplit, a unified framework that integrates three complementary components: EA for quantifying client heterogeneity via evidential statistics, BCC for bias correction through client pairing, and DTD for aligning client and server models to support local inference via dual-level knowledge distillation.

- Extensive experiments on five benchmark datasets demonstrate that BESplit consistently improves effectiveness, efficiency, and convergence stability, outperforming state-of-the-art FL and SFL baselines under heterogeneous non-IID data.

## 2. Related Work

### 2.1. Advances in Split Federated Learning

The decentralized nature of SFL inherently exposes it to both system and data heterogeneity (Thapa et al., 2022; Tirana et al., 2025). To alleviate system heterogeneity in edge environments, MergeSFL (Liao et al., 2024a) improves training efficiency by integrating feature merging and adaptive batch-size regulation, while ParallelSFL (Liao et al., 2024b) adopts a cluster-based training strategy that partitions heterogeneous edge workers and assigns adaptive updating frequencies. HASFL (Lin et al., 2026) further improves convergence efficiency by jointly optimizing batch size and model split. Hourglass (He et al., 2025) focuses

on scalability by enabling multi-GPU data-parallel training with shared model partition, thereby accelerating convergence and improving accuracy. However, while recent studies have made progress in mitigating system heterogeneity, the challenges of data heterogeneity under non-IID settings remain insufficiently explored in SFL.

### 2.2. Federated Learning under Non-IID Data

To mitigate the adverse effects of non-IID data in FL, prior studies have primarily followed two research directions: optimization-based and representation-based approaches.

Optimization-based methods mitigate client drift and stabilize convergence through refined update regularization. FedProx (Li et al., 2020) introduces a proximal term to constrain local objectives, while FedDyn (Acar et al., 2021) further applies dynamic regularization to adaptively balance local optimization and global convergence. In contrast, representation-based methods alleviate distributional bias and enhance feature consistency across clients. Fed-MoE (Jiang et al., 2025) iteratively refines server MoE experts and the gating network, activating relevant experts and promoting diversity to handle non-IID client data. FedRDN (Yan et al., 2025) refines local augmentation with global feature statistics, while FAVD (Kumar et al., 2025) introduces a privacy-preserving, auditable data valuation framework to fairly quantify client contributions during aggregation.

While effective, existing FL methods overlook the structural potential of smashed data. In SFL, the server's access to rich intermediate representations facilitates implicit client coordination, enabling bias-compensation mechanisms that effectively quantify and mitigate client heterogeneity under non-IID conditions.

## 3. Preliminary

### 3.1. Problem Statement

We consider a vanilla SFL system with $K$ clients $\{C_k\}_{k=1}^K$, each holding a private dataset $\mathcal{D}_k \sim P_k$, where $P_k$ denotes the local data distribution. For each client, $\mathcal{D}_k$ is divided into a training set $\mathcal{D}_k^{\text{tr}} = \{(x_j, y_j)\}_{j=1}^{n_k^{\text{tr}}}$ and a test set $\mathcal{D}_k^{\text{te}} = \{(x_j, y_j)\}_{j=1}^{n_k^{\text{te}}}$. In practice, the local distributions $P_k$ vary across clients and deviate from the global distribution $P_g$, resulting in non-IID characteristics in SFL.

Each client model is split into a client-side model $W_{C_k}$ and a server-side model $W_{S_k}$, forming $W_k = [W_{C_k}, W_{S_k}]$. The client-side model maps inputs $x_j$ to smashed features $z_j = W_{C_k}(x_j)$, which are transmitted to the server along with the corresponding labels for further processing. The main server aggregates server-side models $W_{S_g} = A(\{W_{S_k}\}_{k=1}^K)$, while the fed server aggregates client-side models $W_{C_g} = A(\{W_{C_k}\}_{k=1}^K)$, yielding the global model

$W_g = [W_{C_g}, W_{S_g}]$.

Functionally, the client-side model $W_C$ serves as a feature extractor $W_{Ce}$ that maps raw inputs to smashed data, while the server-side model $W_S = [W_{Sp}, W_{Sh}]$ processes these representations through the feature processor $W_{Sp}$ and outputs predictions via the classification head $W_{Sh}$.

**Objective.** The first objective is to learn a global model $W_g$ that generalizes effectively across diverse heterogeneous client distributions:

$$W_g^* = \arg\min_{W_g} \mathbb{E}_{(x,y)\sim P_g}\big[\ell(f_{W_g}(x), y)\big], \quad (1)$$

where $f_{W_g}(\cdot)$ denotes the forward function of $W_g$, and $\ell(\cdot, \cdot)$ is a task-specific loss.

The second objective is to optimize a lightweight auxiliary model $W_A$ for efficient client-side inference, aiming to achieve on-device prediction while maintaining comparable task performance to $W_g$:

$$W_A^* = \arg\min_{W_A} \mathbb{E}_{(x,y)\sim P_g}\big[\ell(f_{W_A}(x), y)\big], \quad (2)$$

### 3.2. Dirichlet-based EDL Framework for SFL

**Model Architecture.** To capture uncertainty arising from client data heterogeneity in SFL, BESplit leverages a Dirichlet-based EDL framework (Sensoy et al., 2018). Each client model $W_k$ produces a class-wise evidence vector $\mathbf{e} = \phi(f(x; W_k))$, where $\phi(\cdot)$ is a non-negative activation function. This yields Dirichlet parameters $\boldsymbol{\alpha} = \mathbf{e} + \mathbf{1}$, which define a Dirichlet distribution over class probabilities and encode both predictive belief and uncertainty. More details on the Dirichlet-based EDL framework are provided in Appendix B. Building on Subjective Logic (Jsang, 2018) and Dempster–Shafer theory (Dempster, 1968; Shafer, 2020), BESplit further decomposes uncertainty into fine-grained, complementary *aleatoric* (Xie et al., 2023) and *epistemic* (Shen et al., 2023) components.

Aleatoric uncertainty captures the inherent ambiguity of the data and is quantified as the expected categorical entropy under the Dirichlet distribution:

$$\begin{aligned} U_{\text{ale}}^{(k)}(x, W_k) &= \mathbb{E}_{Dir(p|x, W_k)}[\mathcal{H}(P(y \mid p))] \\ &= \sum_{i=1}^N \frac{\alpha_i}{S}\big[\psi(S+1) - \psi(\alpha_i + 1)\big], \end{aligned} \quad (3)$$

where $\mathcal{H}(\cdot)$ denotes the Shannon entropy, $Dir(p \mid x, W_k)$ represents the Dirichlet distribution over class probability vector $p$ induced by input $x$ and model $W_k$, $P(y \mid p)$ is the categorical distribution parameterized by $p$, $\psi(\cdot)$ denotes the digamma function, and $S = \sum_i \alpha_i$ is the Dirichlet strength.

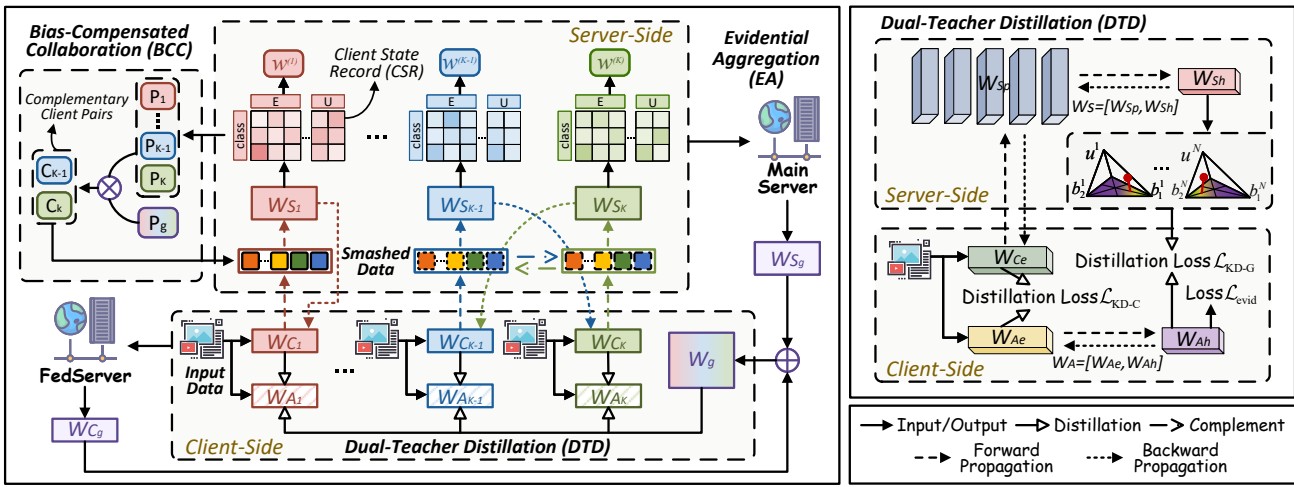

*Figure 1.* Illustration of BESplit. EA, BCC, and DTD modules are integrated into the SFL workflow. In each communication round, client $C_k$ processes a mini-batch of local samples through $W_{C_k}$ to produce smashed data, which are sent to the corresponding $W_{S_k}$ for evidential prediction. The server records these outputs in a client-specific CSR, capturing fine-grained prediction statistics. Based on CSR updates, EA adaptively adjusts the aggregation weights $w^{(k)}$. Then, the server estimates both the global data distribution $P_g$ and each client's local distribution $P_k$ from CSR entries, and activates BCC mechanism to align subsets of smashed features between clients with complementary distributions (e.g., $C_{K-1}$ and $C_K$). Meanwhile, each client maintains an auxiliary model $W_{A_k}$ via DTD, using $W_{C_k}$ and the global model $W_g$ as dual teachers.

Epistemic uncertainty reflects model uncertainty, measured as the differential entropy of the Dirichlet distribution:

$$U_{\text{epi}}^{(k)}(x, W_k) = \mathcal{H}[Dir(p|x, W_k)]$$
$$= \sum_{i=1}^{N} \log \frac{\Gamma(\alpha_i)}{\Gamma(S)} - (\alpha_i - 1)[\psi(\alpha_i) - \psi(S)], \quad (4)$$

where $\Gamma(\cdot)$ is the Gamma function.

**Model Optimization.** To implement EDL, we adopt the standard evidential learning loss, which includes a Kullback–Leibler (KL) regularizer to suppress misleading evidence from incorrect classes:

$$\mathcal{L}_{\text{evid}}(\boldsymbol{\alpha}, \mathbf{y}) = \sum_{i=1}^{N} y_i [\psi(S) - \psi(\alpha_i)]$$
$$+ \lambda_t \, KL\Big[Dir(\mathbf{p}|\tilde{\boldsymbol{\alpha}}) \,\|\, Dir(\mathbf{p}|\mathbf{1})\Big], \quad (5)$$

where $\mathbf{y}$ is the one-hot representation of the ground-truth label, $\tilde{\boldsymbol{\alpha}} = \mathbf{y} + (\mathbf{1} - \mathbf{y}) \odot \boldsymbol{\alpha}$ denotes the adjusted Dirichlet parameters, and $\lambda_t = \min(1.0, t/T)$ is an annealing factor.

## 4. Methodology

### 4.1. Evidential Aggregation

In SFL, non-IID client data often leads to imbalanced contributions during global aggregation. Conventional performance metrics (e.g., accuracy) may overemphasize confident and biased clients, thereby skewing the global update (Nguyen et al., 2015; Pei et al., 2024). To address this,

we design EA, which builds upon the Dirichlet-based EDL framework (Sec. 3.2) to leverage evidential statistics for interpretable and uncertainty-aware client weighting.

**Client State Record.** During communication, the server tracks per-class evidential statistics for each client $k$, as summarized in Table 6 in Appendix C. At round $t$, it collects local evidence and uncertainty statistics to form $\mathbf{R}_{t_c}$ and updates the stored record $\mathbf{R}$ via a staleness-aware exponential moving average (EMA):

$$\mathbf{R} = \beta_{\text{dec}} \, \mathbf{R}_{t_k} + (1 - \beta_{\text{dec}}) \, \mathbf{R}_{t_c}, \quad (6)$$

where $\mathbf{R}_{t_k}$ denotes the latest statistics from the previous participation round $t_k$, and $\beta_{\text{dec}} = \beta^{\max(1, t_c - t_k)}$ is the staleness-aware decay factor.

After the update, the server computes per-class averages by normalizing each statistic with its corresponding sample counts, forming $\bar{\mathbf{R}} = \{\bar{\mathbf{E}}, \mathbf{M}, \bar{\mathbf{U}}_{\text{ale}}, \bar{\mathbf{U}}_{\text{epi}}\}$:

$$\bar{\mathbf{E}} = \text{diag}(\mathbf{M})^{-1}\mathbf{E}, \quad \bar{\mathbf{U}}_{\text{ale}} = \mathbf{U}_{\text{ale}}/\mathbf{M}, \quad \bar{\mathbf{U}}_{\text{epi}} = \mathbf{U}_{\text{epi}}/\mathbf{M}. \quad (7)$$

Where $\mathbf{E}$ denotes the per-class aggregated evidence obtained by accumulating instance-level evidence vectors $\mathbf{e}$ over the client's local samples, and $\mathbf{M}$ records the corresponding per-class sample counts. These normalized statistics, derived from the Dirichlet-based EDL framework, characterize each client's average predictive evidence ($\bar{\mathbf{E}}$) and uncertainty ($\bar{\mathbf{U}}_{\text{ale}}, \bar{\mathbf{U}}_{\text{epi}}$). They are further used to construct two complementary weighting factors: evidence concentration and uncertainty penalization.

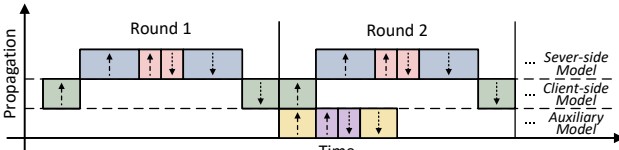

*Figure 2.* SFL process pipeline with parallel knowledge distillation. *Note:* The color scheme of the components is consistent with that used in Fig. 1.

**Evidence Concentration.** To favor clients providing reliable class evidence, the server measures the concentration of evidence on correct predictions via $\bar{\mathbf{E}}^{(k)}$:

$$q_i^{(k)} = \frac{\bar{E}_{i,i}^{(k)}}{\sum_{j=1}^{N} \bar{E}_{i,j}^{(k)} + \epsilon}, \quad Q^{(k)} = \frac{1}{N} \sum_{i=1}^{N} q_i^{(k)}, \quad (8)$$

where $\bar{E}_{i,j}^{(k)}$ denotes the evidence assigned to class $j$ for samples with ground-truth label $i$, and $\epsilon > 0$ ensures numerical stability. A higher $Q^{(k)}$ indicates more concentrated and consistent evidence across classes.

**Uncertainty Penalization.** Clients with high aleatoric or epistemic uncertainty should contribute less to the global update. Thus, the server computes relative uncertainty ratios for each client:

$$R_{\text{ale}}^{(k)} = \frac{\sum_{j=1}^{K} \sum_{i=1}^{N} \bar{U}_{\text{ale},i}^{(j)}}{\sum_{i=1}^{N} \bar{U}_{\text{ale},i}^{(k)} + \epsilon}, \quad R_{\text{epi}}^{(k)} = \frac{\sum_{j=1}^{K} \sum_{i=1}^{N} \bar{U}_{\text{epi},i}^{(j)}}{\sum_{i=1}^{N} \bar{U}_{\text{epi},i}^{(k)} + \epsilon}.$$
$$(9)$$

These ratios penalize clients whose evidential predictions exhibit high uncertainty.

**Client Contribution.** Finally, each client's contribution weight integrates evidence concentration and uncertainty penalization:

$$s^{(k)} = Q^{(k)} \cdot R_{\text{ale}}^{(k)} \cdot R_{\text{epi}}^{(k)}, \quad w^{(k)} = \frac{s^{(k)}}{\sum_{j=1}^{K} s^{(j)}}. \quad (10)$$

This evidential weighting is designed to promote interpretable and robust aggregation by adaptively prioritizing client updates with higher predictive confidence, thereby mitigating the impact of biased local data.

### 4.2. Bias-Compensated Collaboration

To mitigate feature-level non-IID effects, we propose the BCC mechanism, which transforms distributional skew into mutually corrective interactions among clients. Inherently designed as an opportunistic, non-blocking mechanism rather than a synchronous waiting protocol, BCC efficiently operates in a practical streaming environment by

decoupling heavy intermediate activations from lightweight CSRs. Rather than relying on "blind" pairings of asynchronous arrivals, the server continuously maintains the memory-bounded CSRs via an EMA (as described in Section 4.1), thereby preserving a persistent, low-cost approximation of the global distribution. Leveraging this global view, the server evaluates the active incoming stream to perform CSR-guided opportunistic matching on-the-fly, dynamically pairing the most complementary mini-batches.

First, the server quantifies each client's deviation from the global label distribution using the maintained CSR. Clients with significant divergence are identified as biased and grouped as

$$\mathcal{B} = \{C_k \mid d(P_k, P_g) > d_{(i^*)}\}, \quad (11)$$

where $d(\cdot, \cdot)$ denotes the Jensen–Shannon divergence, and the adaptive threshold $d_{(i^*)}$ is defined by the largest consecutive gap in the sorted divergences: $i^* = \arg\max_i \left( d_{(i+1)} - d_{(i)} \right)$, with $d_{(1)} \leq \cdots \leq d_{(K)}$.

Instead of down-weighting or excluding biased clients, BCC collaboratively leverages their complementary distributions to reduce bias. For any two clients $C_i, C_j \in \mathcal{B}$, their distributional complementarity is defined as:

$$\begin{aligned} \mathcal{N}_{i \to j} &= \{(z, y) \in \mathcal{D}_i \mid y \in \mathcal{C}_{i \to j}\}, \\ \mathcal{N}_{j \to i} &= \{(z, y) \in \mathcal{D}_j \mid y \in \mathcal{C}_{j \to i}\}. \end{aligned} \quad (12)$$

where $\| \cdot \|_1$ denotes the $\ell_1$ norm.

To exploit such complementarity, the server constructs a weighted graph over $\mathcal{B}$ with edge weights $\mathrm{G}(C_i, C_j)$, and applies a greedy matching algorithm (Gupta, 2024) to obtain a matching set $M$ of complementary client pairs. For each matched pair $(C_i, C_j) \in M$, the server identifies overrepresented classes:

$$\mathcal{C}_{i \to j} = \{n \mid p_{i,n} > p_{j,n}\}, \quad \mathcal{C}_{j \to i} = \{n \mid p_{j,n} > p_{i,n}\}, \quad (13)$$

where and $\mathcal{C}_{i \to j}$ and $\mathcal{C}_{j \to i}$ correspond to the sets of classes overrepresented in $C_i$ and $C_j$, respectively.

After identifying these classes, the server extracts the corresponding smashed samples:

$$\begin{aligned} \mathcal{N}_{i \to j} &= \{(z, y) \in \mathcal{D}_i \mid y \in \mathcal{C}_{i \to j}\}, \\ \mathcal{N}_{j \to i} &= \{(z, y) \in \mathcal{D}_j \mid y \in \mathcal{C}_{j \to i}\}. \end{aligned} \quad (14)$$

Subsequently, for each matched pair $(C_i, C_j) \in M$, BCC leverages up to $\rho\%$ of complementary class-wise feature signals to refine local representations, where $\rho$ is determined by the theoretical analysis in Appendix D.2. The resulting gradients are aggregated and exclusively routed back to the originating client, thereby mitigating the risk of cross-client information leakage. This interaction induces a coordinated

*Table 2.* Comprehensive overview of the experimental setup.

| Dataset | Total Clients | Clients selected per round | Non-IID Dirichlet parameter $\kappa$ | Model | Epochs | Batch Size |
|---|---|---|---|---|---|---|
| ISIC | 32 | 32 | | DenseNet121 | 100 | 32 |
| HAM10000 | 64 | 64 | 0.1, 0.5, | DenseNet121 | 200 | 32 |
| F-MNIST | 100 | 40 | 1 (Default), | ResNet18 | 300 | 32 |
| CIFAR10 | 100 | 70 | 5, 10 | ResNet18 | 300 | 64 |
| CIFAR100 | 1000 | 200 | | ResNet50 | 500 | 64 |

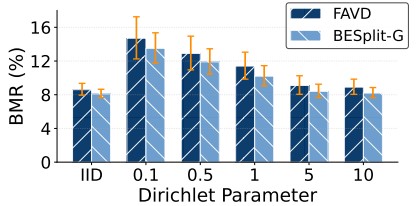

*Figure 3.* BMR comparison under varying data heterogeneity on HAM10000.

optimization signal that approximates training under a more balanced feature distribution.

It is worth noting that BCC introduces no negative effects even if a suitable complementary client is momentarily unavailable under extreme conditions. In such cases, the system does not wait; instead, BCC is simply skipped for that batch, seamlessly falling back to standard EA-based aggregation to ensure stable and continuous updates. Furthermore, the operational memory overhead remains exceptionally minimal and transient. The server is only required to hold two mini-batches simultaneously, resulting in a peak cost of $\mathcal{O}(B \cdot d)$ that is immediately released after the backward pass, where $B$ denotes the batch size and $d$ represents the dimensionality of the smashed data. Because this overhead is bounded purely by the server's concurrent processing capacity rather than the total client population, BCC scales efficiently without imposing noticeable memory or latency burdens compared to standard SFL.

### 4.3. Dual-Teacher Distillation

To enable local inference while mitigating model divergence, we propose DTD, a distillation mechanism tailored to the split architecture of SFL. DTD allows each client to leverage its local data while remaining aligned with the global model through dual-teacher guidance. Fig. 2 demonstrates that the auxiliary model is trained in parallel with SFL, allowing distillation to proceed concurrently without introducing additional communication overhead.

Specifically, each client maintains a lightweight auxiliary model, denoted as $W_A = [W_{Ae}, W_{Ah}]$. Under dual-teacher distillation, the feature extractor $W_{Ae}$ is aligned with the client-side model $W_C$ to preserve local knowledge, while the classification head $W_{Ah}$ is aligned with the global model $W_g$ to capture cross-client patterns. During inference, predictions can be generated using either $W_A$ or $W_g$. In low-connectivity scenarios, $W_A$ enables fully local inference without online servers or full model downloads, facilitating deployment under heterogeneous resource constraints.

**Evidential-Level Distillation with Global Model.** To capture global predictive patterns, $W_A$ is trained to mimic $W_g$ at the evidential level, aligning their predictive distributions to enhance cross-client consistency under heterogeneous data. The evidential distillation loss is defined as

$$\mathcal{L}_{\text{KD-G}} = \tau^2 \text{KL}\big[\boldsymbol{p_a}(\tau) \,\|\, \boldsymbol{p_g}(\tau)\big]$$
$$= \tau^2 \sum_{n=1}^{N} p_g^n(\tau)\Big(\log p_g^n(\tau) - \log p_a^n(\tau)\Big), \quad (15)$$

where $\boldsymbol{p}_a(\tau)$ and $\boldsymbol{p}_g(\tau)$ are the class probability distributions obtained by temperature scaling with $\tau$ applied to the predictions of $W_A$ and $W_g$, respectively.

**Feature-Level Distillation with the Client-Side Model.** To preserve client-specific representations, we align the pairwise relational matrices of batch features derived from $W_C$ and $W_{Ae}$, thereby enforcing structural consistency in the learned feature space while facilitating knowledge transfer.

The pairwise relational matrices are defined using Euclidean distances between batch features:

$$M_C^{ab} = \|z_C^a - z_C^b\|_2^2, \quad M_{Ae}^{ab} = \|z_{Ae}^a - z_{Ae}^b\|_2^2, \quad (16)$$

where $z_C^a$ and $z_{Ae}^a$ denote the $a$-th smashed data produced by $W_C$ and $W_{Ae}$, respectively, and $a, b \in \{1, \ldots, B\}$ index samples within a batch of size $B$. Accordingly, $\boldsymbol{M}_C, \boldsymbol{M}_{Ae} \in \mathbb{R}^{B \times B}$ capture the pairwise relational structures among smashed samples within the batch.

The auxiliary feature extractor is optimized to align the relational structures with the client-side model:

$$\mathcal{L}_{\text{KD-C}} = \tau^2 \text{KL}\Big(\text{LogSoftmax}(\boldsymbol{M}_C/\tau), \text{Softmax}(\boldsymbol{M}_{Ae}/\tau)\Big),$$
$$(17)$$

**Overall Loss.** The total loss of the auxiliary model combines supervised EDL, feature-level distillation, and evidence-level distillation:

$$\mathcal{L}_{\text{total}} = \mathcal{L}_{\text{evid}} + \lambda_C \mathcal{L}_{\text{KD-C}} + \lambda_G \mathcal{L}_{\text{KD-G}}, \quad (18)$$

**Convergence Analysis.** We establish theoretical convergence guarantees for the proposed approach, with detailed derivations presented in Appendix D.

*Table 3.* Performance comparison of BESplit-G and BESplit-A against baselines across datasets under Uniform and Non-IID settings ($\kappa = 1$). **Bold** and underlined numbers indicate the best and second-best results, respectively. Results for $\kappa = 0.1$ and $\kappa = 0.5$ are reported in Appendix F.1.

| Dataset → | ISIC | | HAM10000 | | F-MNIST | | CIFAR10 | | CIFAR100 | |
|---|---|---|---|---|---|---|---|---|---|---|
| SFL Setting → | $n=32, k=32$ | | $n=64, k=64$ | | $n=100, k=40$ | | $n=100, k=70$ | | $n=1000, k=200$ | |
| Method ↓ | Uniform | NIID | Uniform | NIID | Uniform | NIID | Uniform | NIID | Uniform | NIID |
| FedAvg | $69.86_{\pm0.46}$ | $60.28_{\pm1.93}$ | $73.57_{\pm0.37}$ | $63.31_{\pm0.63}$ | $81.83_{\pm0.91}$ | $71.10_{\pm1.34}$ | $62.57_{\pm1.15}$ | $50.55_{\pm1.32}$ | $67.35_{\pm0.68}$ | $62.29_{\pm1.34}$ |
| FedProx | $69.28_{\pm0.53}$ | $60.39_{\pm1.74}$ | $73.17_{\pm0.71}$ | $63.32_{\pm1.21}$ | $81.63_{\pm1.50}$ | $71.21_{\pm1.71}$ | $62.05_{\pm1.40}$ | $50.71_{\pm1.25}$ | $66.49_{\pm0.74}$ | $61.72_{\pm1.58}$ |
| FedDyn | $70.63_{\pm0.42}$ | $62.30_{\pm1.39}$ | $74.74_{\pm0.89}$ | $64.51_{\pm0.72}$ | $82.61_{\pm1.13}$ | $72.35_{\pm1.29}$ | $63.82_{\pm0.91}$ | $54.02_{\pm1.33}$ | $70.97_{\pm0.48}$ | $63.66_{\pm1.19}$ |
| Fed-MoE | $72.45_{\pm0.92}$ | $63.52_{\pm1.40}$ | $75.24_{\pm1.48}$ | $66.76_{\pm1.01}$ | $83.59_{\pm0.86}$ | $72.12_{\pm1.01}$ | $65.13_{\pm0.99}$ | $54.79_{\pm1.07}$ | $71.33_{\pm0.84}$ | $65.31_{\pm1.14}$ |
| FedRDN | $73.58_{\pm0.63}$ | $64.11_{\pm1.37}$ | $76.18_{\pm1.21}$ | $67.23_{\pm0.52}$ | $83.46_{\pm1.56}$ | $73.28_{\pm1.58}$ | $66.14_{\pm1.11}$ | $56.89_{\pm0.94}$ | $72.18_{\pm1.10}$ | $66.28_{\pm0.83}$ |
| FAVD | $73.99_{\pm0.76}$ | $64.52_{\pm0.99}$ | $76.42_{\pm1.54}$ | $67.90_{\pm0.85}$ | $83.72_{\pm1.24}$ | $73.48_{\pm1.73}$ | $66.38_{\pm1.12}$ | $56.13_{\pm1.11}$ | $\underline{72.77}_{\pm1.42}$ | $\underline{66.60}_{\pm1.26}$ |
| SplitFed | $69.02_{\pm1.10}$ | $60.07_{\pm1.21}$ | $72.95_{\pm1.44}$ | $63.76_{\pm1.27}$ | $81.29_{\pm1.28}$ | $71.80_{\pm1.61}$ | $62.31_{\pm1.13}$ | $50.21_{\pm0.91}$ | $67.29_{\pm0.65}$ | $61.64_{\pm0.96}$ |
| MergeSFL | $70.58_{\pm0.67}$ | $62.81_{\pm1.29}$ | $74.21_{\pm0.80}$ | $65.50_{\pm0.86}$ | $82.50_{\pm0.99}$ | $72.62_{\pm1.29}$ | $64.26_{\pm0.98}$ | $53.16_{\pm1.67}$ | $71.47_{\pm1.12}$ | $64.76_{\pm1.45}$ |
| ParallelSFL | $70.59_{\pm0.74}$ | $62.32_{\pm0.95}$ | $72.96_{\pm0.67}$ | $65.07_{\pm1.18}$ | $82.13_{\pm1.21}$ | $72.48_{\pm1.26}$ | $64.33_{\pm0.81}$ | $53.60_{\pm1.15}$ | $70.94_{\pm1.26}$ | $64.84_{\pm1.19}$ |
| HASFL | $70.55_{\pm0.82}$ | $62.52_{\pm1.00}$ | $73.20_{\pm1.19}$ | $65.79_{\pm0.90}$ | $82.71_{\pm.03}$ | $72.43_{\pm1.20}$ | $64.17_{\pm1.21}$ | $53.60_{\pm1.65}$ | $71.17_{\pm1.25}$ | $64.14_{\pm1.02}$ |
| **BESplit-G (ours)** | $\mathbf{75.26}_{\pm0.58}$ | $\mathbf{67.45}_{\pm1.29}$ | $\mathbf{78.51}_{\pm0.96}$ | $\mathbf{70.46}_{\pm1.10}$ | $\mathbf{84.45}_{\pm1.33}$ | $\mathbf{74.89}_{\pm1.66}$ | $\mathbf{68.47}_{\pm1.29}$ | $\mathbf{59.82}_{\pm1.46}$ | $\mathbf{73.19}_{\pm1.20}$ | $\mathbf{68.17}_{\pm0.94}$ |
| **BESplit-A (ours)** | $\underline{73.23}_{\pm0.65}$ | $\underline{66.97}_{\pm1.63}$ | $\underline{76.93}_{\pm1.78}$ | $\underline{68.65}_{\pm1.48}$ | $\underline{83.74}_{\pm1.41}$ | $\underline{74.11}_{\pm1.43}$ | $\underline{67.22}_{\pm1.84}$ | $\underline{57.98}_{\pm1.37}$ | $71.44_{\pm1.51}$ | $66.15_{\pm1.89}$ |

*Figure 4.* Accuracy comparison of BESplit-G and BESplit-A under different non-IID levels.

## 5. Experiments

### 5.1. Experimental Setup

**Data.** We evaluate BESplit on five widely used datasets, including two medical imaging datasets, ISIC (Ogier du Terrail et al., 2022) and HAM10000 (Tschandl et al., 2018), and three natural image datasets, F-MNIST (Xiao et al., 2017), CIFAR10, and CIFAR100 (Krizhevsky et al., 2009). Detailed dataset descriptions are provided in Appendix E.3. To simulate heterogeneous client distributions, we use a Dirichlet-based partitioning scheme, where the concentration parameter $\kappa$ controls the degree of non-IID skew across clients (see Appendix E.2 for details).

**Baselines.** BESplit jointly optimizes two complementary objectives: BESplit-G (EA + BCC), which optimizes the global model, and BESplit-A (EA + BCC + DTD), which optimizes local auxiliary models for on-device inference. We evaluate the effectiveness of both BESplit-G and BESplit-A against a comprehensive set of approaches addressing data heterogeneity. This includes several advanced methods, such as FedAvg (McMahan et al., 2017), FedProx, and FedDyn, as well as recent state-of-the-art (SOTA) methods, including Fed-MoE, FedRDN, and FAVD. In addition, we compare BESplit and BESplit-A with the latest SFL ap-

proaches, namely SplitFed (Thapa et al., 2022), MergeSFL, ParallelSFL, and HASFL. Detailed descriptions of all baselines are provided in Appendix E.1.

**Evaluation Metrics.** To comprehensively evaluate the performance of BESplit, we employ four metrics: (1) Test Accuracy (ACC), reflecting the overall predictive performance; (2) Round-to-Accuracy (RTA), the number of communication rounds to achieve a predefined target accuracy; (3) Time-to-Accuracy (TTA), the total wall-clock time required to reach a target accuracy; and (4) Benign Misclassification Rate (BMR), a metric specifically used in medical applications to assess model reliability and safety by measuring the rate of clinically non-benign misclassifications.

**Implementation Details.** All experiments were conducted on a single NVIDIA A100 GPU with 40 GB of memory. By default, the SGD optimizer was employed for all image classification datasets with a learning rate of $1 \times 10^{-4}$. For the DTD stage, the distillation temperature was set to $\tau = 5$, while the weighting coefficients were configured as $\lambda_C = 0.2$ and $\lambda_G = 0.3$, following the hyperparameter settings detailed in Section 5.5. The primary experimental configurations for each dataset are summarized in Table 2.

*Table 4.* Results of ablation study across five datasets.

| Component | ISIC | HAM10000 | F-MNIST | CIFAR10 | CIFAR100 |
|---|---|---|---|---|---|
| **BESplit** | **67.45**$_{\pm 1.29}$ | **70.46**$_{\pm 1.10}$ | **74.89**$_{\pm 1.66}$ | **59.82**$_{\pm 1.46}$ | **68.17**$_{\pm 0.94}$ |
| w/o EA | 64.34$_{\pm 1.02}$ | 67.58$_{\pm 1.05}$ | 73.04$_{\pm 1.04}$ | 57.49$_{\pm 0.80}$ | 65.52$_{\pm 0.89}$ |
| w/o E | 65.36$_{\pm 0.85}$ | 68.15$_{\pm 0.29}$ | 73.47$_{\pm 0.34}$ | 57.93$_{\pm 0.25}$ | 66.42$_{\pm 0.63}$ |
| w/o $U_{ale}$ | 66.84$_{\pm 0.63}$ | 68.96$_{\pm 0.34}$ | 74.38$_{\pm 0.46}$ | 58.96$_{\pm 0.80}$ | 67.23$_{\pm 0.64}$ |
| w/o $U_{epi}$ | 66.54$_{\pm 0.62}$ | 69.27$_{\pm 0.73}$ | 74.11$_{\pm 0.45}$ | 59.24$_{\pm 0.52}$ | 67.75$_{\pm 0.41}$ |
| w/o BCC | 63.59$_{\pm 1.32}$ | 66.46$_{\pm 1.48}$ | 72.14$_{\pm 1.12}$ | 55.65$_{\pm 1.44}$ | 64.28$_{\pm 0.97}$ |

*Table 5.* Comparison of RTA on the CIFAR100 dataset. Each value represents the number of communication rounds required to achieve the target accuracy.

| Target Acc. | FedAvg | | FAVD | | MergeSFL | | BESplit | |
|---|---|---|---|---|---|---|---|---|
| | Uniform | NIID | Uniform | NIID | Uniform | NIID | Uniform | NIID |
| 15% | 16$_{\pm 2}$ | 36$_{\pm 3}$ | 14$_{\pm 2}$ | 30$_{\pm 3}$ | 15$_{\pm 1}$ | 33$_{\pm 2}$ | **13**$_{\pm 1}$ | **29**$_{\pm 2}$ |
| 30% | 49$_{\pm 2}$ | 102$_{\pm 5}$ | 39$_{\pm 3}$ | 88$_{\pm 4}$ | 44$_{\pm 3}$ | 92$_{\pm 4}$ | **35**$_{\pm 2}$ | **79**$_{\pm 3}$ |
| 45% | 106$_{\pm 4}$ | 179$_{\pm 7}$ | 91$_{\pm 5}$ | 153$_{\pm 6}$ | 100$_{\pm 3}$ | 169$_{\pm 5}$ | **83**$_{\pm 3}$ | **134**$_{\pm 4}$ |
| 60% | 282$_{\pm 8}$ | 475$_{\pm 12}$ | 257$_{\pm 7}$ | 401$_{\pm 10}$ | 268$_{\pm 6}$ | 419$_{\pm 9}$ | **247**$_{\pm 6}$ | **371**$_{\pm 11}$ |

## 5.2. Comprehensive Performance and Robustness

In this section, we evaluate BESplit on five datasets against ten state-of-the-art baselines under varying levels of data heterogeneity.

BESplit-G consistently outperforms all baseline models under non-IID conditions, achieving the best performance across all datasets. As shown in Table 3, it exceeds classical FL algorithms, recent SOTA FL approaches, and SFL counterparts, demonstrating superior adaptability to heterogeneous data. On average, BESplit-G achieves a 3.0% higher test accuracy than the best baseline under default non-IID settings, with the largest gains on the medical datasets ISIC and HAM10000, underscoring its suitability for sensitive medical imaging tasks.

To assess its robustness under increasingly skewed data distributions, we evaluate BESplit-G and representative baselines across varying non-IID intensities. As illustrated in Fig. 4, BESplit-G consistently outperforms the baselines as the Dirichlet parameter decreases, especially under extremely non-IID settings, confirming its adaptability and increasing advantage under severe data heterogeneity. We further assess its reliability using the BMR metric under varying levels of data heterogeneity on the HAM10000 dataset. As shown in Fig. 3, BESplit-G consistently achieves lower BMR values than the strongest baseline, demonstrating stronger robustness and dependability in safety-critical medical prediction scenarios.

BESplit-A achieves competitive performance while supporting fully local inference, making it well suited for resource-constrained client devices. As shown in Table 3 and Fig. 4, this lightweight auxiliary model, whose compact architecture contains far fewer parameters than the global model (as detailed in Appendix F.4), preserves local knowledge and effectively mitigates the impact of data heterogeneity. Across varying non-IID intensities, BESplit-A consistently

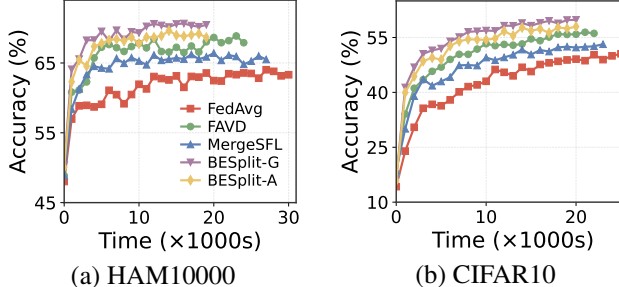

(a) HAM10000      (b) CIFAR10

*Figure 5.* Comparison of TTA on the HAM10000 and CIFAR10 datasets.

maintains strong performance, achieving 1.3% higher accuracy than the best baseline under the default non-IID setting and performing closely to BESplit-G, demonstrating its effectiveness and deployability in practical SFL systems.

## 5.3. Ablation Study

We conduct ablation studies on five datasets to assess the contribution of each component in BESplit, with results presented in Table 4. Replacing EA with simple average aggregation or removing any EA submodule consistently degrades global alignment, indicating that EA stabilizes training and ensures more consistent global updates under non-IID data. The largest performance drop occurs when BCC is removed, underscoring its critical role in mitigating the effects of heterogeneous data distributions. Overall, these results indicate that BESplit's robustness arises from the joint effect of EA and BCC.

## 5.4. Efficiency Analysis

**Computational Cost.** Computational cost is evaluated using the TTA metric, which measures the total wall-clock time required to reach the target accuracy. As shown in Fig. 5, BESplit consistently reaches the target accuracy faster than all baselines. Specifically, on the HAM10000 and CIFAR10 datasets, BESplit-G achieves 1.58× and 1.25× speedup over FedAvg, respectively, while also outperforming the strongest baseline in computation cost and achieving higher under heterogeneous SFL scenarios. These results validate the lightweight design of BEsplit, which effectively reduces computational overhead, enabling more scalable and efficient SFL training.

**Convergence.** We evaluate the convergence efficiency of BESplit using the Round-to-Accuracy (RTA) metric. A lower RTA indicates faster convergence and, under comparable per-round transmission costs, higher communication efficiency. As shown in Table 5, BESplit consistently achieves the target accuracy in fewer rounds than all baselines on the CIFAR100 dataset. While SOTA baselines exhibit similar trends, they converge more slowly under non-IID condi-

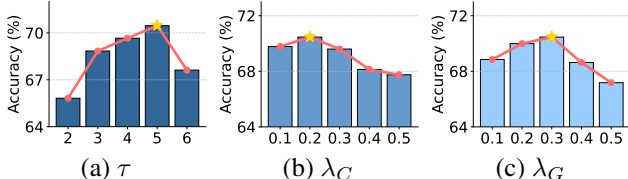

*Figure 6.* Sensitivity analysis of BESplit on the F-MNIST dataset.

tions, and the gap widens as the accuracy target increases, reflecting accumulated optimization bias. In contrast, BE-Split maintains stable improvement throughout training and achieves the lowest RTA across all targets, with particularly pronounced gains under non-IID scenarios. This demonstrates that BESplit achieves fast and stable convergence, effectively mitigating the impact of heterogeneous data.

### 5.5. Hyperparameter Sensitivity

We conduct a sensitivity analysis of BESplit on the F-MNIST dataset with respect to three key hyperparameters: the distillation temperature $\tau$, and the regularization coefficients $\lambda_C$ and $\lambda_G$. As illustrated in Fig. 6, BESplit consistently achieves stable performance across a wide and practically relevant range of values for all three parameters, exhibiting only minor fluctuations in accuracy. This suggests that BESplit is not overly sensitive to hyperparameter selection within reasonable settings, highlighting its robustness and practical reliability under non-IID conditions.

## 6. Discussion

Our framework combines three complementary mechanisms: EA, BCC, and DTD to systematically address data heterogeneity in SFL. EA allows clients to compute adaptive aggregation weights based on per-class evidence and uncertainty, improving robustness under non-IID conditions. BCC reduces distributional skew by leveraging complementary feature signals between biased clients, thereby mitigating class imbalance and enhancing convergence stability. DTD enables the auxiliary model to integrate knowledge from both client-side and global models while preserving evidential supervision, providing a principled approach to knowledge transfer.

In terms of privacy, although BESplit operates under a standard honest-but-curious server threat model, we acknowledge that the server's access to smashed data and labels carries inherent feature-inversion risks common to all SFL architectures. Nevertheless, our proposed CSRs introduce no additional instance-level attack surface or raw feature leakage. Instead, they only maintain coarse-grained, aggregated class-level statistics with minimal storage overhead (i.e., $\mathcal{O}(K \times N^2)$). Furthermore, to ensure that historical information gradually decays and to prevent long-term data

retention, the memory-bounded CSRs utilize an EMA. In addition, a Time-To-Live (TTL) mechanism is applied to automatically discard statistics from inactive clients. Ultimately, BESplit strictly preserves the core federated mandate that raw sensitive data never leaves the local device. While building a formal defense against all reconstruction attacks remains an open challenge in SFL, the practical risks within BESplit could be naturally mitigated by server-side non-linearities, implicit loss mixing, and the bounded ratio $\rho$. Integrating Differential Privacy (DP) to further mask these aggregated statistics presents a promising, orthogonal direction for future work.

The framework also presents some limitations. First, the effectiveness of BCC relies on the estimation of distributional deviations and complementary pairings, and inaccuracies may affect the quality of guidance provided by complementary signals. Second, DTD benefits from appropriately selected distillation coefficients and temperature parameters to balance feature-level, evidential-level, and supervised losses, which may require adaptation to different datasets. Finally, the approach assumes partially complementary client data distributions, and performance may be influenced under extreme or highly heterogeneous conditions. Future work could investigate automated hyperparameter tuning and methods to enhance robustness in more challenging scenarios.

## 7. Conclusion

In this work, we tackle the challenge of non-IID data in Split Federated Learning (SFL). Unlike prior methods, we explicitly exploit the split architecture to simultaneously address data heterogeneity at the aggregation, feature, and model levels. The proposed BESplit framework unifies EA, BCC, and DTD into a principled and robust solution for SFL under non-IID conditions. Extensive experiments on five benchmark datasets show that BESplit achieves higher accuracy and faster convergence than state-of-the-art FL and SFL baselines, demonstrating superior effectiveness, scalability, and practical deployability.

## Acknowledgment

This work was supported by the National Key R&D Program of China (2023YFA1009500).

## Impact Statement

This paper presents work whose goal is to advance the field of Machine Learning. There are many potential societal consequences of our work, none which we feel must be specifically highlighted here.

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

# A. Related Work

### A.1. Uncertainty-aware Federated Learning

Uncertainty (Han et al., 2022) is generally categorized into two primary forms: aleatoric uncertainty (Xie et al., 2023), which arises from inherent data noise, and epistemic uncertainty (Shen et al., 2023), which stems from incomplete knowledge of the underlying model. Accurate quantification of both forms is essential for assessing the reliability and trustworthiness of deep learning systems (Pei et al., 2024; Liu et al., 2025).

In FL, uncertainty has been increasingly leveraged to enhance robustness, reliability, and decision-making. For instance, RIPFL (Qin et al., 2023) employs Dempster–Shafer theory for interpretable client selection and Bayesian evidence for robust model aggregation, while Delphi (Aristodemou et al., 2025) explores parameter perturbation to expose potential trustworthiness vulnerabilities. Among uncertainty estimation methods, EDL (Sensoy et al., 2018) is a representative framework that utilizes deep neural networks (DNNs) (Xu et al., 2024; Yu et al., 2024a) to quantify predictive evidence. Building upon this foundation, recent studies have integrated EDL into FL to address heterogeneous and safety-critical scenarios. For example, RESFL (Wasif et al., 2025) combines adversarial privacy disentanglement with fairness-aware aggregation for autonomous vehicle object detection. FedEvi (Chen et al., 2024a) adopts Dirichlet-based EDL to disentangle epistemic and aleatoric uncertainties, thereby improving aggregation for medical image segmentation. Similarly, FEAL (Chen et al., 2024b) incorporates Dirichlet-based EDL into federated active learning (Kim et al., 2023), enabling joint modeling of both uncertainty types under domain shifts in medical applications.

Despite these advances, most existing EDL-based methods are designed specifically for conventional FL settings, with limited exploration in SFL. Given the unique architectural and optimization challenges of SFL, directly applying these approaches is non-trivial. This gap underscores the need for uncertainty-aware methods tailored to SFL to further improve its robustness and reliability.

### A.2. Knowledge Distillation

Knowledge distillation (KD) (Sun et al., 2024; Muralidharan et al., 2024; Liu et al., 2024) trains a compact student model to imitate a teacher model, enabling effective knowledge transfer for model compression. It has recently been extended to SFL to address system and data heterogeneity (Zhang et al., 2024). For example, FedGKT (He et al., 2020) enables bidirectional knowledge transfer between edge and server models, integrating FL and SL while maintaining lightweight edge computation. FSMKD (Luo & Zhang, 2024) further introduces mutual distillation between personalized local models and a shared global model. Similarly, Li et al. (Li et al., 2024) align an auxiliary model with the main model to improve robustness under varying communication conditions.

In contrast to prior approaches that perform distillation within a single dataset (Muralidharan et al., 2024; Yang et al., 2024), rely on a single large teacher model (Liu et al., 2024; Pham et al., 2024), or assume multiple models sharing a common backbone (Luo & Zhang, 2024), BESplit allows each client to fully exploit its local data while aligning with the global model. This design ensures consistent performance across heterogeneous clients and better mitigates the challenges of highly non-IID data in SFL.

# B. Dirichlet-based Evidential Deep Learning for Uncertainty Modeling

To enable interpretable fine-grained aggregation in SFL, we adopt a Dirichlet-based EDL framework grounded in evidential learning theory (Sensoy et al., 2018). This framework quantifies predictive uncertainty and improves the robustness of aggregation under heterogeneous non-IID client distributions (Qu et al., 2024).

For an $N$-class classification task, the client's full model $W_k$ maps an input $x$ to logits $f(x, W_k)$, which parameterize a Dirichlet distribution over the class probabilities $\boldsymbol{p}$, thereby capturing second-order uncertainty:

$$Dir(\boldsymbol{p} \mid \boldsymbol{\alpha}) = \begin{cases} \dfrac{1}{B(\boldsymbol{\alpha})} \displaystyle\prod_{i=1}^{N} p_i^{\alpha_i - 1}, & \boldsymbol{p} \in \Delta^N, \\ 0, & \text{otherwise}, \end{cases} \tag{19}$$

where $\Delta^N = \{\boldsymbol{p} \mid \sum_{i=1}^{N} p_i = 1, \ 0 < p_i < 1\}$ denotes the probability simplex, $B(\boldsymbol{\alpha})$ is the Beta function, and

$\boldsymbol{\alpha} = [\alpha_1, \ldots, \alpha_N]^\top$ are the Dirichlet parameters.

The Dirichlet parameters $\boldsymbol{\alpha}$ are obtained from non-negative evidence $\mathbf{e} = [e_1, \ldots, e_N]^\top$ via a non-negative activation $\phi(\cdot)$:

$$\mathbf{e} = \phi(f(z, W_S)) \geq 0, \quad \boldsymbol{\alpha} = \mathbf{e} + \mathbf{1}, \quad S = \sum_{i=1}^N \alpha_i, \tag{20}$$

where $S$ is the Dirichlet strength.

Following Subjective Logic (Jsang, 2018) and Dempster-Shafer theory (Dempster, 1968; Shafer, 2020), the Dirichlet distribution is mapped to belief and uncertainty measures:

$$\boldsymbol{b} = \frac{\mathbf{e}}{S}, \quad u = \frac{N}{S}, \quad \mathbb{E}[p_i] = b_i + \frac{u}{N} = \frac{\alpha_i}{S}, \tag{21}$$

where $\boldsymbol{b} = [b_1, \ldots, b_N]^\top$ represents the belief mass, and $u$ quantifies evidential uncertainty.

## C. Evidential Statistics

During each communication round, the server maintains class-wise evidential statistics for each client, as summarized in Table 6.

*Table 6.* Class-wise statistics for each client in BESplit.

| Statistic | Symbol | Description |
|---|---|---|
| Evidence Matrix | $\mathbf{E} \in \mathbb{R}^{N \times N}$ | evidence per class |
| Sample Count Vector | $\mathbf{M} \in \mathbb{R}^N$ | number of samples per class |
| Aleatoric Uncertainty Vector | $\mathbf{U}_{\text{ale}} \in \mathbb{R}^N$ | uncertainty from data per class |
| Epistemic Uncertainty Vector | $\mathbf{U}_{\text{epi}} \in \mathbb{R}^N$ | uncertainty from model per class |

## D. Theoretical Analysis

### D.1. Server-side Parameter Deviation Analysis

**Assumption D.1.** For each class $n \in \{1, \ldots, N\}$, the class-wise loss function $\mathcal{L}_n(W_S) = \mathbb{E}_{z|y=n}[\ell(f_{W_S}(z), y)]$ is $\beta_n$-smooth with respect to the server-side model parameters $W_S$, i.e

$$\|\nabla\mathcal{L}_n(W_{S_1}) - \nabla\mathcal{L}_n(W_{S_2})\| \leq \beta_n \|W_{S_1} - W_{S_2}\|.$$

**Assumption D.2.** There exists a constant $g_{\max} > 0$ such that, for all classes $n \in \{1, \ldots, N\}$ and for all server-side model parameters $W_S$, the norm of the class-wise gradient is uniformly bounded, i.e.,

$$\|\nabla\mathcal{L}_n(W_S)\| \leq g_{\max}.$$

**Assumption D.3.** We treat the empirical class proportion $P_k$ as an unbiased estimator of the underlying true label distribution. In the SFL setting, clients typically possess sufficiently large local datasets, rendering the estimation error negligible. At communication round $t$, the server aggregates server-side model from $K$ clients. The aggregated server-side gradient is $g_t = \sum_{k=1}^K w^{(k)} g_k$, where $w^{(k)}$ denotes the aggregation weight for client $k$. Conditioned on $W_{S_g}^{(t-1)}$, the expected aggregated gradient $\bar{g}_t = \mathbb{E}[g_t \mid W_{S_g}^{(t-1)}]$ can be decomposed into the global gradient and a bias term:

$$\bar{g}_t = \nabla\mathcal{L}_g(W_{S_g}^{(t-1)}) + b_t,$$

where $\nabla\mathcal{L}_g = \sum_{n=1}^N P_{g,n} \nabla\mathcal{L}_n(W_{S_g}^{(t-1)})$ is the global gradient and $b_t$ captures the bias induced by heterogeneity in client label distributions.

Moreover, the stochastic variance of the aggregated gradient is uniformly bounded:

$$\mathbb{E}\left\|g_t - \bar{g}_t\right\|^2 \le \sigma^2.$$

**Lemma D.4** (Gradient Bias Bound). *Under Assumption D.2, the gradient bias $b_t$ is bound as:*

$$\|b_t\| = \|\bar{g}_t - \nabla\mathcal{L}(W_{S_g}^{(t-1)})\| \le \delta_{BCC} := g_{max} \sum_{k=1}^{K} w^{(k)} \|P_k - P_g\|_1. \tag{22}$$

*Proof.*

$$\|\bar{g}_t - \nabla\mathcal{L}(W_{S_g}^{(t-1)})\| = \left\|\sum_{k=1}^{K} w^{(k)} \sum_{n=1}^{N} (P_{k,n} - P_{g,n})\nabla\mathcal{L}_n(W_{S_g}^{(t-1)})\right\| \tag{23}$$
$$\le \sum_{k=1}^{K} w^{(k)} \|P_k - P_g\|_1 \max_n \|\nabla\mathcal{L}_n(W_{S_g}^{(t-1)})\|.$$

Applying Assumption D.2, the result $\|\bar{g}_t - \nabla\mathcal{L}\| \le g_{max} \sum w^{(k)} \|P_k - P_g\|_1$ holds. $\square$

**Lemma D.5** (Server-side Parameter Deviation). *Let $W_{S_c}^{(t)}$ denote the server-side parameter after the $t$-th update in the centralized setting. Under Assumptions D.1, the deviation between $W_{S_g}^{(t)}$ and $W_{S_c}^{(t)}$ satisfies*

$$\mathbb{E}\|W_{S_g}^{(t)} - W_{S_c}^{(t)}\| \le (1 + \eta L)\mathbb{E}\|W_{S_g}^{(t-1)} - W_{S_c}^{(t-1)}\| + \eta\delta_{BCC} + \eta\sigma. \tag{24}$$

*Proof.* From the update rules $W_{S_g}^{(t)} = W_{S_g}^{(t-1)} - \eta g_t$ and $W_{S_c}^{(t)} = W_{S_c}^{(t-1)} - \eta\nabla\mathcal{L}(W_{S_c}^{(t-1)})$, we have

$$W_{S_g}^{(t)} - W_{S_c}^{(t)} = (W_{S_g}^{(t-1)} - W_{S_c}^{(t-1)}) - \eta\big(g_t - \nabla\mathcal{L}(W_{S_c}^{(t-1)})\big). \tag{25}$$

Taking norms and applying the triangle inequality yields

$$\|W_{S_g}^{(t)} - W_{S_c}^{(t)}\| \le \|W_{S_g}^{(t-1)} - W_{S_c}^{(t-1)}\| + \eta\|g_t - \nabla\mathcal{L}(W_{S_c}^{(t-1)})\|. \tag{26}$$

Taking expectation and inserting intermediate gradients, we obtain

$$\mathbb{E}\|g_t - \nabla\mathcal{L}(W_{S_c}^{(t-1)})\| \le \mathbb{E}\|g_t - \bar{g}_t\| + \|\bar{g}_t - \nabla\mathcal{L}(W_{S_g}^{(t-1)})\| + \|\nabla\mathcal{L}(W_{S_g}^{(t-1)}) - \nabla\mathcal{L}(W_{S_c}^{(t-1)})\|. \tag{27}$$

By Assumption D.3, $\mathbb{E}\|g_t - \bar{g}_t\| \le \sigma$. The second term corresponds to the distributional bias controlled by BCC and is bounded by $\delta_{BCC}$. By $L$-smoothness,

$$\|\nabla\mathcal{L}(W_{S_g}^{(t-1)}) - \nabla\mathcal{L}(W_{S_c}^{(t-1)})\| \le L\|W_{S_g}^{(t-1)} - W_{S_c}^{(t-1)}\|. \tag{28}$$

Substituting the above bounds completes the proof. $\square$

*Remark* D.6 (Motivation of BCC). Lemma D.4 establishes a rigorous theoretical basis showing that the server-side gradient deviation $b_t$ is strictly upper-bounded by the weighted $\ell_1$ divergence between local and global label distributions. Specifically, the inequality $|b_t| \le g_{max} \sum w^{(k)} |P_k - P_g|_1$ reveals that the discrepancy between the aggregated surrogate gradient and the true global gradient arises directly from local label skew. Consequently, reducing the distributional mismatch $|P_k - P_g|_1$ provides a mathematically grounded strategy for mitigating gradient drift.

Motivated by this observation, BCC leverages complementary label distributions among biased clients to mitigate skew. Rather than discarding or down-weighting divergent clients, it pairs those with complementary overrepresented classes and redistributes a controlled fraction of their feature signals. Executed on the server, this alignment reduces overall label skew without extra communication, thereby lowering server-side gradient bias and fostering more balanced representations.

**D.2. Guidance for Ratio $\rho^*$**

For a matched client pair $(C_i, C_j) \in M$, the objective of BCC is to minimize their joint contribution to the global gradient bias induced by label distribution mismatch. We focus on a single class $n$ and, without loss of generality, assume $P_{i,n} \geq P_{j,n}$. The class-wise bias minimization problem is formulated as

$$\min_{\rho} \mathcal{F}_{i,j,n}(\rho) = w_t^{(i)} |\hat{P}_{i,n}(\rho) - P_{g,n}| + w_t^{(j)} |\hat{P}_{j,n}(\rho) - P_{g,n}|. \tag{29}$$

At communication round $t$, the empirical label proportion is modeled as $P_{k,n}^{(t)} = P_{k,n} + \varepsilon_{k,n}^{(t)}$, where $\varepsilon_{k,n}^{(t)}$ denotes zero-mean sampling noise. Accordingly, the historical estimate $P_k$ serves as a consistent and low-variance approximation of the underlying client distribution. Moreover, since the adaptive aggregation weights $w_t^{(k)}$ are determined after the server-side update, we approximate them using their values from the previous round, i.e., $w_t^{(k)} \approx w_{t-1}^{(k)}$.

Under the BCC update rule, a fraction $\rho P_{i,n}$ of class-$n$ mass is transferred from client $i$ to client $j$, yielding $\hat{P}_{i,n} = P_{i,n} - \rho P_{i,n}$ and $\hat{P}_{j,n} = P_{j,n} + \rho P_{i,n}$. Substituting these expressions into the objective leads to

$$\min_{\rho} \mathcal{F}_{i,j,n}(\rho) = w_{t-1}^{(i)} |P_{i,n} - \rho P_{i,n} - P_{g,n}| + w_{t-1}^{(j)} |P_{j,n} + \rho P_{i,n} - P_{g,n}|. \tag{30}$$

1. **Case 1:** $w^{(i)} > w^{(j)}$. When client $i$ is assigned higher reliability, minimizing the joint bias prioritizes aligning its effective label proportion with the global distribution, resulting in

$$\rho_{i,j,n}^* = \frac{P_{i,n} - P_{g,n}}{P_{i,n}}. \tag{31}$$

2. **Case 2:** $w^{(j)} > w^{(i)}$. When client $j$ is more reliable, the minimum is achieved by fully compensating its deviation from the global distribution, yielding

$$\rho_{i,j,n}^* = \frac{P_{g,n} - P_{j,n}}{P_{i,n}}. \tag{32}$$

3. **Case 3:** $w^{(k)} = w^{(j)}$. If both clients are equally weighted, the objective is flat within a feasible interval, and any

$$\left[ \frac{\min(P_{i,n} - P_{g,n}, P_{g,n} - P_{j,n})}{P_{i,n}}, \frac{\max(P_{i,n} - P_{g,n}, P_{g,n} - P_{j,n}))}{P_{i,n}} \right]$$

achieves the same minimal joint bias.

Following the derivation of the optimal complementarity ratios $\rho_{i,j,n}^*$ for each matched client pair, we next formalize the practical implementation of BCC. For any candidate pair $(i, j)$, we define the edge weight in the bipartite matching graph as

$$G(C_i, C_j) = \|P_i - P_g\|_1 + \|P_j - P_g\|_1 - \|(P_i - P_g) + (P_j - P_g)\|_1. \tag{33}$$

This quantity measures the potential reduction in total $\ell_1$ deviation when clients $i$ and $j$ are jointly compensated. Only pairs with complementary deviations from $P_g$ yield positive weights, while aligned or negligible deviations result in zero weight and are thus excluded from matching.

For analytical simplicity, we consider uniform aggregation weights $w^{(k)} = 1/K$. Under this setting, the greedy algorithm prioritizes client pairs that maximize the reduction in the cumulative label deviation $\sum_k \|P_k - P_g\|_1$. By systematically shrinking global label imbalance, BESpit effectively controls the induced gradient bias $\delta_{\text{BCC}}$, leading to more stable and well-conditioned server-side optimization.

**D.3. Theoretical Guarantees for Convergence**

We now establish that BESplit converges to a stationary point under non-IID conditions.

**Theorem D.7.** *Let the learning rate satisfy $\eta_t = \eta \leq \frac{1}{4L}$. Under Assumption D.1 and D.3, the sequence of server-side models $\{W_{S_g}^{(t)}\}_{t=0}^{T-1}$ generated according to*

$$W_{S_g}^{(t)} = W_{S_g}^{(t-1)} - \eta g_t,$$

*with $g_t = \sum_{k=1}^{K} w^{(k)} g_k$ denoting the aggregated gradient and $w^{(k)}$ denoting the aggregation weights determined by the EA mechanism , satisfies*

$$\min_{t \in \{0,\ldots,T-1\}} \mathbb{E}[\|\nabla\mathcal{L}(W_{S_g}^{(t)})\|^2] \leq \frac{4(\mathcal{L}(W_{S_g}^{(0)}) - \mathcal{L}_*)}{\eta T} + 4\delta_{BCC}^2 + 4L\eta\sigma^2, \tag{34}$$

*where $\mathcal{L}_*$ denotes the global minimum.*

**Proof.** From the $L$-smoothness of $\mathcal{L}$ and the update rule $W_{S_g}^{(t)} = W_{S_g}^{(t-1)} - \eta g_t$, we have:

$$\mathbb{E}[\mathcal{L}(W_{S_g}^{(t)})] \leq \mathbb{E}[\mathcal{L}(W_{S_g}^{(t-1)})] - \eta\langle\nabla\mathcal{L}(W_{S_g}^{(t-1)}), \bar{g}_t\rangle + \frac{L\eta^2}{2}\mathbb{E}[\|g_t\|^2]. \tag{35}$$

The first-order term is bounded using Young's inequality:

$$-\langle\nabla\mathcal{L}(W_{S_g}^{(t-1)}), \bar{g}_t\rangle \leq -\frac{1}{2}\|\nabla\mathcal{L}(W_{S_g}^{(t-1)})\|^2 + \frac{1}{2}\delta_{BCC}^2. \tag{36}$$

For the second-order term, we decompose the second moment into bias and variance:

$$\mathbb{E}[\|g_t\|^2] = \|\bar{g}_t\|^2 + \mathbb{E}[\|g_t - \bar{g}_t\|^2] \leq 2\|\nabla\mathcal{L}(W_{S_g}^{(t-1)})\|^2 + 2\delta_{BCC}^2 + \sigma^2. \tag{37}$$

Substituting these bounds into Eq.(35) gives:

$$\begin{aligned}
\mathbb{E}[\mathcal{L}(W_{S_g}^{(t)})] &\leq \mathbb{E}[\mathcal{L}(W_{S_g}^{(t-1)})] - \frac{\eta}{2}\|\nabla\mathcal{L}(W_{S_g}^{(t-1)})\|^2 + \frac{\eta}{2}\delta_{BCC}^2 + L\eta^2\|\nabla\mathcal{L}(W_{S_g}^{(t-1)})\|^2 + L\eta^2\delta_{BCC}^2 + \frac{L\eta^2\sigma^2}{2} \\
&\leq \mathbb{E}[\mathcal{L}(W_{S_g}^{(t-1)})] - (\frac{\eta}{2} - L\eta^2)\|\nabla\mathcal{L}(W_{S_g}^{(t-1)})\|^2 + (\frac{\eta}{2} + L\eta^2)\delta_{BCC}^2 + \frac{L\eta^2\sigma^2}{2}.
\end{aligned} \tag{38}$$

Since $\eta \leq \frac{1}{4L}$, it follows that $\frac{\eta}{2} - L\eta^2 \geq \frac{\eta}{4}$ and $\frac{\eta}{2} + L\eta^2 \leq \frac{3\eta}{4}$. We have

$$\mathbb{E}[\mathcal{L}(W_{S_g}^{(t)})] \leq \mathbb{E}[\mathcal{L}(W_{S_g}^{(t-1)})] - \frac{\eta}{4}\|\nabla\mathcal{L}(W_{S_g}^{(t-1)})\|^2 + \frac{3\eta}{4}\delta_{BCC}^2 + \frac{L\eta^2\sigma^2}{2}. \tag{39}$$

Summing the above inequality from $t = 1$ to $T$ yields

$$\begin{aligned}
\mathbb{E}[\mathcal{L}(W_{S_g}^{(T)})] \leq{}& \mathcal{L}(W_{S_g}^{(0)}) - \frac{\eta}{4}\sum_{t=0}^{T-1}\mathbb{E}[\|\nabla\mathcal{L}(W_{S_g}^{(t)})\|^2] \\
&+ \frac{3\eta T}{4}\delta_{BCC}^2 + \frac{L\eta^2 T}{2}\sigma^2.
\end{aligned} \tag{40}$$

Using the lower boundedness of $\mathcal{L}$ and rearranging terms, we obtain

$$\frac{1}{T}\sum_{t=0}^{T-1}\mathbb{E}[\|\nabla\mathcal{L}(W_{S_g}^{(t)})\|^2] \leq \frac{4(\mathcal{L}(W_{S_g}^{(0)}) - \mathcal{L}_*)}{\eta T} + 3\delta_{BCC}^2 + 2L\eta\sigma^2. \tag{41}$$

Consequently,

$$\min_{t \in \{0,\ldots,T-1\}} \mathbb{E}[\|\nabla\mathcal{L}(W_{S_g}^{(t)})\|^2] \leq \frac{4(\mathcal{L}(W_{S_g}^{(0)}) - \mathcal{L}_*)}{\eta T} + 3\delta_{BCC}^2 + 2L\eta\sigma^2. \tag{42}$$

With a constant step size, BESplit converges to a neighborhood of a stationary point. The optimization error decreases at rate $O(1/T)$, whereas the asymptotic error floor is dominated by the label distribution skewness induced bias. In contrast to standard aggregation schemes that incur a large bias under non-IID data, the proposed method explicitly suppresses this bias, leading to a markedly tighter convergence neighborhood.

$\square$

# E. Implementation Details

### E.1. Baseline Details

To provide a comprehensive comparison, we evaluate BESplit against a broad range of representative baselines from both FL and SFL.

**Classical and advanced FL algorithms:**

- **FedAvg** (McMahan et al., 2017) performs local stochastic gradient updates on each client and aggregates the resulting models via weighted averaging on a central server, enabling collaborative training without sharing raw data.

- **FedProx** (Li et al., 2020) extends FedAvg to handle both system and statistical heterogeneity by adding a proximal term to each client's local objective, penalizing deviations from the global model for more stable and accurate convergence.

**Recent state-of-the-art FL approaches:**

- **Fed-MoE** (Jiang et al., 2025) iteratively updates server-side MoE experts and the gating network using a small reserved dataset, selectively activates relevant experts during inference, and enforces expert diversity through routed experts and a gating entropy loss, thereby addressing non-IID client data.

- **FedRDN** (Yan et al., 2025) mitigates feature distribution skew by injecting local dataset statistics into augmented samples, exposing clients to broader data distributions. It is lightweight, network-agnostic, and compatible with standard augmentation pipelines to enhance generalization.

- **FAVD** (Kumar et al., 2025) evaluates client contributions using local and global data density functions with added Gaussian noise for privacy. This enables auditable updates and robustly mitigates the impact of malicious or skewed data.

**Recent SFL methods:**

- **SplitFed** (Thapa et al., 2022) combines FL and SL to enhance privacy and model robustness. By splitting the model between clients and server and incorporating differential privacy and PixelDP, it reduces client computation while maintaining accuracy, suitable for resource-constrained or streaming-data environments.

- **MergeSFL** (Liao et al., 2024a) addresses system and statistical heterogeneity via feature merging and batch-size regulation. Merged features approximate IID mini-batches, while dynamic batch sizes stabilize top-model updates, improving accuracy and training efficiency.

- **ParallelSFL** (Liao et al., 2024b) clusters workers by computing capability and data distribution to reduce waiting time. KL-divergence ensures local data within clusters is near-IID, and diverse update frequencies further enhance convergence and efficiency.

- **HASFL** (Lin et al., 2026) jointly optimizes batch size and model split under resource constraints to balance convergence efficiency, training accuracy, and communication-computation latency in heterogeneous edge environments.

All baselines are implemented or reconfigured according to their official repositories or original descriptions to ensure fair and reproducible comparisons.

## E.2. Non-IID Data Allocation

To simulate realistic federated scenarios, data is distributed among clients using both IID and non-IID schemes.

**IID Allocation:** Each of the $K$ clients receives an equal number of randomly selected samples, ensuring identical local data distributions across clients. This provides a baseline for evaluating federated algorithms under balanced conditions.

**Non-IID Allocation:** Heterogeneous client data is generated via a Dirichlet-based strategy. Let $N$ be the number of classes and $K$ the number of clients. For each class $n$, its samples are shuffled and partitioned among clients according to a Dirichlet distribution with parameter $\alpha$. Smaller $\alpha$ produces more skewed distributions, creating clients with imbalanced or specialized class compositions. Sample counts are rounded and adjusted to ensure all samples are assigned. Formally, the class-$n$ sample proportions for clients are

$$(p_1, \ldots, p_K) \sim \text{Dirichlet}(\alpha, \ldots, \alpha),$$

with corresponding counts corrected to match the total number of class-$n$ samples. This produces realistic non-IID partitions reflecting statistical heterogeneity across clients, a key challenge in SFL.

## E.3. Dataset Details

We evaluate BESplit and the baseline methods on both medical and natural image datasets, covering a range of complexity, class imbalance, and domain heterogeneity to reflect realistic non-IID federated scenarios.

**ISIC (Ogier du Terrail et al., 2022):** Approximately 25,000 RGB dermoscopic images with expert annotations across multiple skin lesion categories, including melanoma, nevus, and keratosis. Variations in acquisition devices and clinical sites introduce strong inter-domain heterogeneity, making it a challenging benchmark for federated and split learning.

**HAM10000 (Tschandl et al., 2018):** 10,015 dermoscopic images across 7 lesion categories with significant class imbalance (115–6,705 samples per class), reflecting realistic clinical heterogeneity in non-IID client distributions.

**Fashion-MNIST (F-MNIST) (Xiao et al., 2017):** 70,000 grayscale images (28×28) across 10 fashion categories, featuring more complex textures and shapes than MNIST while remaining compact for federated experiments.

**CIFAR-10 (Krizhevsky et al., 2009):** 60,000 RGB images (32×32) spanning 10 object classes such as animals and vehicles, suitable for evaluating model generalization under non-IID distributions.

**CIFAR-100 (Krizhevsky et al., 2009):** Extends CIFAR-10 to 100 fine-grained classes (600 images per class), with higher intra-class variance and inter-class similarity, offering a more challenging testbed for scalable representation learning in federated settings.

# F. Additional Experiments

We present supplementary experiments that extend and complement the empirical analyses in the main text.

## F.1. Comparative Evaluation Under Severe Non-IID Distributions

To systematically assess the robustness of BESplit under highly heterogeneous data, we conducted experiments with Dirichlet parameters of $\kappa = 0.1$ and $\kappa = 0.5$, compared against ten state-of-the-art baseline models across five datasets. This setup evaluates how BESplit behaves as data heterogeneity increases, reflecting challenging real-world scenarios. Detailed results are reported in Table 7.

Both BESplit variants, BESplit-G and BESplit-A, consistently outperform classical FL methods and recent state-of-the-art FL or SFL approaches across all datasets and heterogeneity levels, highlighting their robustness against performance degradation under non-IID distributions. Importantly, the relative advantage of BESplit grows as data heterogeneity intensifies. This pattern indicates that the framework not only withstands the challenges posed by skewed client distributions but actively leverages the collaborative split architecture to maintain stable learning. In extreme non-IID scenarios, where many baseline methods experience significant performance drops, BESplit-G continues to achieve substantial improvements, demonstrating strong adaptability and resilience. BESplit-A, while more compact and fully supporting local inference, preserves a substantial portion of this advantage, effectively mitigating heterogeneity effects and providing a practical solution for resource-constrained client devices.

*Table 7.* Performance comparison of BESplit-G and BESplit-A against baseline methods across multiple datasets under severe non-IID conditions (with heterogeneity levels $\kappa = 0.1$ and $\kappa = 0.5$). **Bold** and underlined numbers indicate the best and second-best results, respectively.

| Dataset → | ISIC | | HAM10000 | | F-MNIST | | CIFAR10 | | CIFAR100 | |
|---|---|---|---|---|---|---|---|---|---|---|
| **SFL Setting →** | $n$=32, $k$=32 | | $n$=64, $k$=64 | | $n$=100, $k$=40 | | $n$=100, $k$=70 | | $n$=1000, $k$=200 | |
| **Method ↓** | $\kappa = 0.1$ | $\kappa = 0.5$ | $\kappa = 0.1$ | $\kappa = 0.5$ | $\kappa = 0.1$ | $\kappa = 0.5$ | $\kappa = 0.1$ | $\kappa = 0.5$ | $\kappa = 0.1$ | $\kappa = 0.5$ |
| FedAvg | 36.19 | 50.21 | 46.08 | 56.63 | 55.10 | 64.27 | 29.19 | 42.77 | 57.97 | 59.80 |
| FedProx | 38.56 | 51.60 | 46.70 | 57.34 | 55.25 | 65.10 | 28.15 | 42.84 | 58.99 | 60.25 |
| FedDyn | 44.27 | 52.12 | 48.30 | 58.82 | 57.72 | 64.98 | 32.92 | 43.42 | 58.25 | 62.32 |
| Fed-MoE | 46.32 | 55.41 | 49.69 | 59.56 | 60.08 | 65.90 | 37.14 | 46.72 | 60.17 | 63.89 |
| FedRDN | 46.24 | 56.12 | 49.21 | 60.23 | 59.50 | 66.15 | 37.31 | 46.52 | 61.75 | 64.93 |
| FAVD | 46.79 | 57.69 | 50.79 | 60.31 | 59.18 | 67.96 | 36.19 | 47.35 | 61.25 | 64.30 |
| SplitFed | 36.82 | 50.32 | 45.88 | 56.34 | 54.52 | 64.65 | 29.54 | 41.99 | 57.96 | 58.86 |
| MergeSFL | 42.74 | 54.38 | 48.74 | 58.25 | 57.78 | 66.63 | 33.83 | 44.97 | 60.80 | 62.86 |
| ParallelSFL | 41.60 | 52.79 | 47.35 | 58.10 | 57.16 | 66.28 | 33.73 | 43.38 | 60.50 | 61.10 |
| HASFL | 42.10 | 53.28 | 48.03 | 57.99 | 57.85 | 66.31 | 33.33 | 44.09 | 60.14 | 61.66 |
| **BESplit-G (ours)** | **52.19** | **61.24** | **56.10** | **65.63** | **62.72** | **69.52** | **42.22** | **51.48** | **63.70** | **66.82** |
| **BESplit-A (ours)** | 49.74 | 59.51 | 53.74 | 62.51 | 60.78 | 67.99 | 40.29 | 49.97 | 62.20 | 64.23 |

## F.2. BMR-Based Reliability Analysis

To further evaluate reliability and robustness, we measured the BMR across varying levels of data heterogeneity on the medical datasets HAM10000 and ISIC, focusing on comparison against the strongest baseline FAVD. BMR captures the consistency and safety of predictions under non-IID data distributions.

As shown in Fig. 7, BESplit-G consistently achieves the lowest BMR across both datasets, indicating superior reliability in medical prediction tasks. Notably, the performance gap between BESplit-G and FAVD widens as data heterogeneity increases: whereas FAVD exhibits substantial deterioration in BMR under extreme non-IID scenarios, BESplit-G maintains stable and low misclassification rates, reflecting strong resilience to skewed client distributions.

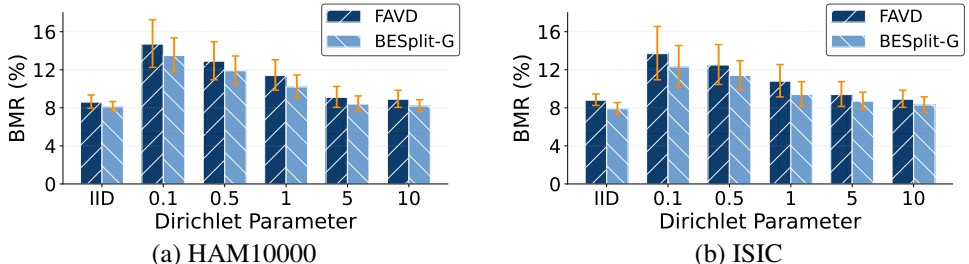

(a) HAM10000           (b) ISIC

*Figure 7.* BMR comparison under varying degrees of data heterogeneity on medical datasets (HAM10000 and ISIC).

## F.3. Robustness Against Extreme Data Heterogeneity

The training behaviors of BESplit-G and BESplit-A under increasing levels of non-IID data heterogeneity are shown in Fig. 8. As data heterogeneity intensifies, both methods exhibit more pronounced fluctuations in their training trajectories, reflecting the substantial impact of uneven label distributions on optimization stability, particularly under extreme skew. BESplit-G maintains comparatively smooth curves and keeps performance variations within a narrow band even at high non-IID levels. This stability arises from its direct access to global knowledge throughout training, which effectively regularizes the updates and suppresses gradient noise.

In contrast, BESplit-A displays larger oscillations. This behavior is primarily attributable to the limited expressive capacity of its lightweight auxiliary model: under severe distribution skew, the model becomes more susceptible to gradient variability, causing the Non-IID–induced noise to be amplified across updates. Although the dual-teacher distillation mechanism alleviates part of this instability by blending both local and global supervisory signals, the resulting trajectories remain

visibly less stable than those of BESplit-G.

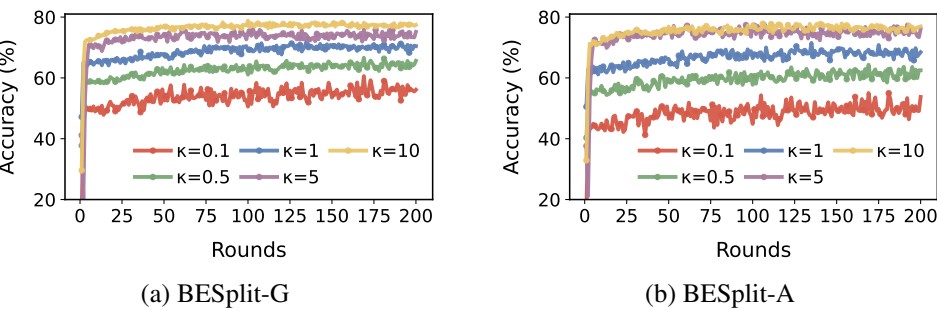

(a) BESplit-G            (b) BESplit-A

*Figure 8.* Training curves of BESplit-G and BESplit-A under varying non-IID level on the HAM10000 dataset.

### F.4. Model Size Evaluation

As shown in Table 8, the auxiliary model is intentionally designed to be highly lightweight. Its parameter scale is only slightly larger than that of the client-side model and remains orders of magnitude smaller than the global model. As a result, maintaining the auxiliary model introduces only a relatively small overhead, reinforcing its practicality for deployment on resource-constrained client devices within BESplit.

*Table 8.* Parameter counts for different model configurations within the BESplit system.

| Backbone | Client-side Model | Auxiliary Model | Global Model |
|---|---|---|---|
| DenseNet121 | 344.5K | 347.1K | 8.0M |
| ResNet18 | 83.6K | 84.2K | 11.7M |
| ResNet50 | 120.1K | 120.7K | 25.5M |

