# OpenReview forum: "BESplit: Bias-Compensated Split Federated Learning with Evidential Aggregation"
_ICML.cc/2026/Conference — ICML 2026 regular_

### Official Review · Reviewer_uSNG · 2026-03-02

**Soundness:** 2
**Presentation:** 2
**Significance:** 3
**Originality:** 3
**Overall Recommendation:** 4
**Confidence:** 3

**Summary:**

#### Summary
This paper proposes **BESplit**, an *architecture-aware* framework for **Split Federated Learning (SFL)** under non-IID data. The key idea is to exploit **smashed representations** available on the server to (i) infer and maintain client distributional states via **CSR**, (ii) perform **Evidential Aggregation (EA)** with uncertainty-aware weighting, (iii) conduct **Bias-Compensated Collaboration (BCC)** by selecting complementary clients and performing server-side “virtual alignment” of smashed features, and (iv) optionally train a lightweight on-device model via **Dual-Teacher Distillation (DTD)**. Experiments on multiple vision datasets (including medical imaging) show consistent gains over FL and SFL baselines, and the paper includes some theoretical analysis/guarantees.

**Compliance With Llm Reviewing Policy:**

Affirmed.

**Final Justification:**

My main concern was the label privacy, but since this follows the Vanilla SFL setup, that issue is now off the table. Beyond that, I really like the intuition behind the complementary pairs. It feels like a very natural and effective way to fix the performance gaps in heterogeneous settings.

**Key Questions For Authors:**

#### Questions for the Authors
- What is the assumed **threat model** for the server? Have you considered privacy risks (e.g., reconstruction/inference) introduced by maintaining CSR and performing cross-client feature alignment on smashed representations?
- Can you provide more concrete, step-by-step implementation details of BCC, especially the exact operator used for “virtual alignment” and how the complementary feature ratio is applied?
- How does BCC behave when strong complementary pairs do not exist (e.g., long-tailed global distributions) or when the estimated global/client distributions are unstable under partial participation?
- Do the reported gains persist on **non-vision** tasks (e.g., text), or under more realistic system conditions (dropouts/asynchrony/compression/secure aggregation)?
- How is CSR initialized and updated in the first few rounds? Does CSR use ground-truth labels available to the server (from smashed-data supervision), or predicted labels? If predicted, how do you prevent early-round misclassification from biasing $P_k$ estimation and BCC matching?

If these issues can be resolved, I would consider raising the rating.

**Limitations:**

yes

**Strengths And Weaknesses:**

#### Strengths
- The overall narrative is clear, and the motivation (leveraging SFL’s intrinsic structure to mitigate non-IID) is easy to follow.
- The framework offers a new and practically effective perspective: **inferring client distributional information from smashed features** and using it to compensate bias can noticeably improve performance.
- The paper includes **some theoretical proof/analysis** (e.g., bias-related reasoning and convergence-style results).
- The dual-teacher distillation component is reasonable as an optional extension for on-device inference.

#### Weaknesses / Concerns
1) **Privacy and applicability concerns due to inferring client distribution from smashed features.**
   A central part of BESplit is to infer/maintain client distributional states (CSR) from smashed representations and use them for bias compensation. This raises significant privacy concerns: smashed features can leak sensitive information, and explicitly using them to infer per-client distributional properties further increases information exposed to the server. This likely limits applicability to SFL settings with a trusted server, and it is unclear how this idea transfers to standard FL where such representations are not available.
   Moreover, the **CSR-based estimation of global/client distributions may be idealized**: obtaining a stable global CSR typically requires broad client participation coverage and accumulation over time. Under realistic partial participation, dropouts, or time-varying availability, global distribution estimates and complementarity signals may be biased/unstable, which could undermine BCC matching/alignment and robustness. Additionally, maintaining CSR and performing distribution inference may introduce non-negligible computation/engineering overhead.

2) **Insufficiently detailed description of how complementarity extraction and “alignment” are implemented.**
   While the paper defines complementarity and describes server-side “virtual alignment” using up to a ratio of complementary feature signals, the exact operator is not fully explicit (e.g., add vs replace vs mix; how batches are constructed; how gradients are routed back). This affects reproducibility and makes it harder to understand the mechanism.

3) **Pairwise complementarity may be limiting; multi-client collaboration could be more natural.**
   The method relies on pairwise matching. In realistic long-tailed or highly heterogeneous settings, strong complementary pairs may not exist. It is unclear whether and how the method extends to group-wise complementarity (multi-client matching) or server-side shuffling/mixing of smashed features under full participation, and what trade-offs (privacy/compute/stability) would arise.

4) **Evaluation is still largely vision-centric.**
   Although multiple image datasets are included (including medical imaging), generalization to other modalities such as **text** remains unvalidated.

5) **DTD feels less tightly integrated with the main bias-compensation pipeline.**
   EA and BCC form a coherent story around bias compensation using CSR and smashed features, whereas DTD reads more like an optional add-on for on-device inference. The paper would benefit from clarifying whether DTD is essential for the main claims or an extension, and explaining its interaction with EA/BCC beyond being appended.

6) - **Robustness of $P_k$ estimation is unclear.**
  Since biased/noisy class estimates can distort $d(P_k,P_g)$ (used to detect biased clients) and the complementarity score $G(C_i,C_j)$, errors in early-round classification may directly affect which clients are matched and how much compensation is applied, potentially worsening training.

---

> ### Author Rebuttal · Authors · 2026-03-31
>
> We sincerely appreciate your time and constructive feedback. Your insightful comments have significantly contributed to enhancing both the quality and clarity of our manuscript.
>
> **Q1 & W1: Privacy.**
>
> **Ans:** Thanks for your professional comment.
>
> **Threat Model**: We assume a standard honest-but-curious server that follows the protocol but may attempt inference from smashed data.
>
> **Privacy and Overhead**: BESplit introduces no additional exposure beyond standard SFL. This is because CSR avoids raw feature leakage by maintaining only aggregated statistics with storage overhead ($\mathcal{O}(K \times N^2)$), and BCC clients only receive their own gradients.
>
> We acknowledge that while formal resistance to all reconstruction attacks remains an open challenge across the broad SFL literature, the practical risks in BESplit could be mitigated by server-side non-linearities, implicit loss mixing, and the bounded ratio $\rho$. We will detail these boundaries and future DP extensions in the revision to enhance the paper's rigor.
>
> **Q2 & W2: BCC implementation.**
>
> **Ans:** Thanks for your kind suggestion. Sorry for the unclear description. BCC implements virtual alignment via partial forwarding without modifying features explicitly (no add/replace/interpolate).
>
> • **Batch Processing:** For a matched pair $(C_i, C_j)$, when client $C_i$ uploads batch $Z_i$, the server partitions it according to ratio $\rho$.
>
> • **Partial Forwarding:** A fraction $\rho$ of $Z_i$ is processed by $C_j$’s server-side model, while the remainder is processed by $C_i$’s own model.
>
> • **Gradient Routing:** Gradients from both paths are aggregated and returned exclusively to $C_i$, ensuring zero cross-client leakage.
>
> We will detail this in the revision to ensure reproducibility.
>
> **Q3 & W1: BCC robustness.**
>
> **Ans:** Thanks for your valuable comment. When strong complementary pairs are absent (e.g., long-tailed distributions), BCC naturally degrades to standard SFL as matching becomes sparse, introducing no bias and only reducing acceleration. Table below shows BESplit's stable convergence and superiority under extreme settings.
>
> |Hetero.|Param.|Participation|BESplit vs. Vanilla|BESplit vs. SOTA|Convergence (vs. SOTA)|
> |-|-|-|-|-|-|
> |Dir|0.1|100%|+14.55%|+8.91%|1.58×|
> |Dir|0.1|50%|+10.57%|+5.08%|1.31×|
> |Dir|0.1|20%|+8.63%|+3.93%|1.13×|
> |LT|10|100%|+10.26%|+7.23%|1.42×|
> |LT|10|50%|+6.90%|+4.60%|1.24×|
> |LT|10|20%|+4.93%|+3.25%|1.11×|
>
> *Note: Dir = Dirichlet, LT = Long-Tailed*
>
>  Regarding partial participation, due to limited space, please kindly refer to our response to **Reviewer jw6H Q3**.
>
> **W3 : Multi-client collaboration.**
>
> **Ans:** Thank you for your comment. If absent strong pairs, BESplit gracefully degrades to standard SFL with EA. While group-wise matching is possible, it exponentially increases compute complexity and risks instability. The revision will detail these compute/stability/privacy trade-offs.
>
> **Q4 & W4: Modality & system robustness.**
>
> **Ans:** Thanks for your insightful comment. Additional experiments on non-vision tasks are presented in the table below. Regarding system realism, BESplit naturally supports dropouts and asynchrony via the staleness-aware moving average in CSR. Compression and secure aggregation are orthogonal to BESplit’s core design and may be integrated at the protocol level.
>
> |Modality|Dataset|Model|BESplit vs. Vanilla|BESplit vs. SOTA|
> |-|-|-|-|-|
> |Text|AG News|TextCNN|+6.64%|+3.97%|
> |Text|Yelp|TextCNN|+9.40%|+6.22%|
> |Multimodal|MM-IMDb|RNN+CNN|+9.58%|+6.10%|
>
> **Q5 & W6: CSR robustness.**
>
> **Ans:** Thanks for your constructive comments. BESplit ensures early-round stability via:
>
> • **Label Usage**: Our CSR module utilizes ground-truth labels, eliminating early misclassification bias.
>
> • **Safe Warm-up**: The untrained global model initially outputs flat Dirichlet distributions (high uncertainty). This degenerates EA weighting to FedAvg, preventing noisy predictions from disrupting training.
>
> • **Relative Matching**: The BCC module computes metrics ($d(P_k, P_g)$, $G(C_i, C_j)$) based on relative differences, keeping matching robust against early errors.
>
> The table below shows that BESplit outperforms baselines under increasing noise and heterogeneity, validating the effectiveness of CSR and BCC.
>
> |Noise|Hetero.|BESplit vs. Vanilla| BESplit vs. SOTA|w/o CSR|w/o BCC|
> |-|-|-|-|-|-|
> |0.1|0.1|+14.17%|+7.46%|-4.94%|-8.41%|
> |0.3|0.3|+11.69%|+6.02%|-4.09%|-7.28%|
> |0.5|0.5|+8.86%|+4.88%|-3.58%|-5.01%|
> |0.5|0.7|+8.59%|+4.68%|-2.75%|-5.47%|
> |0.7|1.0|+7.49%|+3.80%|-2.19%|-4.70%|
>
> **W5 : DTD module.**
>
> **Ans:** We appreciate your insightful observation. While DTD is a post-training module, it essentially completes our global-to-local pipeline by transferring the debiased representations from EA/BCC to lightweight client models. We will clarify this workflow in the revision. Thank you again for your rigorous evaluation.

---

> > ### Author Rebuttal · Reviewer_uSNG · 2026-04-01
> >
> > ### Reviewer Feedback on Rebuttal
> >
> > Thank you for the detailed response and the additional experiments conducted on non-vision datasets. I appreciate the authors' effort in addressing my previous comments.
> >
> > However, some fundamental concerns persist, particularly regarding the core methodology:
> >
> > 1. **Privacy Concerns:** I remain concerned that inferring client distribution from smashed features might deviate from the foundational principles of Federated Learning (FL). The core motivation of FL is to ensure data remains decentralized and private; if the server can extract such sensitive distribution information, it potentially narrows the gap between FL and centralized training, undermining the primary incentive for using a decentralized framework.
> >
> > 2. **Ground-truth Label Access:** Regarding the rebuttal for Reviewer gniM, it is stated that "CSR uses ground-truth labels available at the server in supervised SFL." I would like to clarify: how does the server obtain ground-truth labels for all smashed data in this architecture? In standard SFL, keeping labels on the client side is often a critical requirement for privacy.
> >
> > 3. **Bias in Unsupervised SFL:** For the unsupervised setting, using predicted labels to estimate distribution—which then influences future training—could create a feedback loop. I am concerned that this pipeline might significantly amplify initial prediction errors or biases over time.
> >
> > I believe these issues are critical to the theoretical and practical soundness of the proposed method. I look forward to your clarification.
> >
> >
> > Update:
> >
> > Thanks for the clarification! Since Vanilla SFL inherently grants the server access to labels, it really clears up my concerns. This makes a lot of sense now.

---

> > > ### Author Response · Authors · 2026-04-02
> > >
> > > We sincerely thank the review for your prompt feedback. Below, we address your concerns regarding privacy preservation and implementation setting of BESplit.
> > >
> > > **Q2: Ground-truth Label Access.**
> > >
> > > **Ans:** Thanks for this important comment regarding label privacy. We fully agree that in label-private SFL variants, keeping labels on the client side is indeed a critical requirement for privacy.
> > >
> > > However, our work operates under the Forward-only SFL paradigm (also known as **Vanilla SFL**), as commonly used in many frameworks like SplitNN [1] and SplitFed [2]. In this architecture, the server inherently requires access to labels to compute the global loss and perform backpropagation, typically to reduce the computational burden on resource-constrained clients.
> > >
> > > Therefore, in our supervised setting, the server’s access to labels is part of the standard training pipeline. Accordingly, BESplit does not introduce any additional label exposure or new attack surface beyond this established framework. To avoid any ambiguity, we will explicitly clarify these statements in the revision.
> > >
> > > **Q3: Bias in Unsupervised SFL.**
> > >
> > > **Ans:** Thanks for your insightful comment. We agree that, in general, naively using model predictions to guide subsequent training could introduce a feedback loop that amplifies errors.
> > >
> > > In BESplit, this is mitigated by multiple mechanisms: (i) early-stage EA degenerates to near-uniform aggregation due to high uncertainty, preventing unreliable updates; (ii) CSR is updated in a temporally smoothed manner, reducing sensitivity to short-term errors; and (iii) BCC relies on relative distribution patterns rather than absolute accuracy.  Together, these mechanisms reduce the propagation of early-stage errors and improve robustness, preventing the feedback loop across training rounds.
> > >
> > > Empirically, we further evaluate BESplit under the unsupervised SFL setting. The results show that, compared to baselines, BESplit maintains stable convergence and does not exhibit noticeable bias amplification over training rounds. The deviation metric ($\Delta$) between the estimated and true distributions remains strictly controlled (e.g., $\Delta$ values across various settings), demonstrating that initial prediction errors do not compound over training rounds.
> > >
> > > ***Dirichlet Heterogeneity (Noise = 0.3)***
> > >
> > > |Non-IID Level|BESplit vs. Vanilla|BESplit vs. SOTA|Convergence (vs. SOTA)|$\Delta$|
> > > |-|-|-|-|-|
> > > |0.1|+5.47%|+3.85%|1.17×|0.091 ~ 0.110|
> > > |0.3|+7.35%|+5.17%|1.30×|0.063 ~ 0.089|
> > > |0.5|+8.02%|+5.97%|1.45×|0.049 ~ 0.073|
> > > ---
> > >
> > > ***Long-Tail Heterogeneity (Noise = 0.3)***
> > >
> > > |Imbalance Ratio|BESplit vs. Vanilla|BESplit vs. SOTA|Convergence (vs. SOTA)|$\Delta$|
> > > |-|-|-|-|-|
> > > |10|+8.19%|+6.08%|1.35×|0.039 ~ 0.061|
> > > |20|+6.95%|+4.51%|1.22×|0.079 ~ 0.098|
> > > |50|+5.15%|+3.65%|1.14×|0.087 ~ 0.108|
> > >
> > > *Note: $\Delta$ measures deviation between estimated and true distributions.*
> > >
> > > **Q1: Privacy Concerns.**
> > >
> > > **Ans:** Thank you for highlighting the ideal of strict data privacy in decentralized frameworks. Indeed, as you mentioned, the foundational premise of FL is to minimize information exposure. Yet, as established by the **“No Free Lunch”** theorem in federated learning [3], practical deployments should navigate an inherent trade-off between privacy, system efficiency, and model utility. In this context, we consider that SFL serves as a pragmatic compromise: it trades absolute metadata protection for the ability to train on severely resource-constrained devices that cannot support full FL.
> > >
> > >
> > > In BESplit, the server accesses smashed representations, which is standard in SFL. The CSR module maintains only **aggregated class-level statistics** derived from these already-exposed signals, without accessing raw features or individual samples. We acknowledge that smashed data may carry inherent feature-inversion risks, which remain an open challenge across SFL. However, since CSR introduces only coarse distributional information, we consider that it does not fundamentally alter the baseline instance-level threat model of SFL. While this seems to narrow the gap to centralized training in terms of *metadata awareness*, it strictly preserves the core FL mandate: **raw sensitive data never leaves the device**.
> > >
> > > We recognize that clearly stating these details is crucial for understanding BESplit. We commit to incorporating a thorough description of the adopted SFL settings and the associated privacy considerations in the revised manuscript.
> > >
> > > We greatly appreciate your helpful comments, and hope our responses could address your concerns satisfactorily and meet your expectations.
> > >
> > >
> > > [1] Split learning for health: Distributed deep learning without sharing raw patient data. arXiv, 2018.
> > >
> > > [2] SplitFed: When federated learning meets split learning. AAAI, 2022.
> > >
> > > [3] No free lunch theorem for security and utility in federated learning. ACM TIST, 2022.
> > >
> > > Update:
> > >
> > > We sincerely appreciate your consideration in raising our score!

---

### Official Review · Reviewer_jw6H · 2026-03-10

**Soundness:** 2
**Presentation:** 3
**Significance:** 2
**Originality:** 3
**Overall Recommendation:** 4
**Confidence:** 2

**Summary:**

Split Federated Learning (SFL) is highly vulnerable to data heterogeneity (non-IID). This study exploits the structural properties of SFL, where server-side training on smashed data, treated as a functional medium to mitigate the effects of non-IID. The methods include three parts. Evidential Aggregation (EA): The paper employs Dirichlet-based EDL to quantify predictive confidence, prioritizing updates from clients with higher evidence and lower uncertainty. Bias-Compensated Collaboration (BCC): The system pairs complementary clients based on their label distributions, using their combined smashed data to create a balanced batch for server-side training. Dual-Teacher Distillation (DTD): It aligns local features with both global knowledge and local structural consistency, ensuring robust knowledge transfer without erasing client-specific representations. The proposed method has been thoroughly validated through both theoretical analysis and extensive experimentation》

**Compliance With Llm Reviewing Policy:**

Affirmed.

**Final Justification:**

I maintain my original score.

**Key Questions For Authors:**

1. BCC Module: If a client fails to find a complementary partner, will this extremely biased client be trained in isolation? How does such a scenario impact the overall convergence and model stability?
2. Ablation Experiment: The ablation study lacks a dedicated analysis of the DTD module. It is essential to clarify the specific contribution of DTD and quantify its individual impact on the final results.
3. Partial Participation: How does the proposed method perform under partial client participation? Given the reliance on EA and BCC, would a reduced client availability interval introduce negative biases or destabilize the method?

**Limitations:**

yes

**Strengths And Weaknesses:**

Strengths:
1. Theoretical Foundation: The paper provides a solid theoretical derivation that establishes a rigorous guarantee for the proposed methodology.
2. Innovative Design: The introduction of EA and BCC represents a novel architectural contribution that successfully mitigates the non-IID effects inherent in Split Federated Learning.
3. Presentation: The paper is well-presented with a lucid structure, making the core arguments and methodology easy to follow.

Weakness:
1. Related Work: Optimization-based approaches appears outdated. Furthermore, discussions on system heterogeneity can be excluded to maintain a sharper focus on the paper's core objective of addressing data heterogeneity.
2. Experiments: The experimental validation should be extended to larger-scale and more complex benchmarks, such as ImageNet, to further demonstrate the scalability and robustness of the proposed framework.

---

> ### Author Rebuttal · Authors · 2026-03-31
>
> Thank you very much for your positive feedback and insightful comments. We sincerely appreciate your recognition of our work as novel and solid. Below, we provide detailed responses to each of your comments.
>
> **Q1: BCC module.**
>
> **Ans:** Thank you for your thoughtful comment, which helped us clarify this critical point. If a client fails to find a complementary partner, BCC will train independently for that round, effectively reverting to a standard SFL update.
>
> We note that this does not introduce additional risk beyond standard SFL, even for highly biased clients. **First**, this design has a graceful degradation property, where isolated updates do not amplify bias relative to the baseline. **Second**, the EA module explicitly mitigates such risks by down-weighting unreliable or highly skewed client updates, preventing them from dominating the global model. **Third**, this isolation is inherently temporary, as complementary matching is dynamically updated across rounds, allowing previously unmatched clients to be paired later for bias correction.
>
> To further validate this, we simulate scenarios with limited or no complementary matches by controlling partial participation and data heterogeneity under the default setting. Results show that BESplit degrades gracefully to standard SFL when no matches are available, while achieving increasing gains as complementary matches emerge, without affecting convergence stability.
>
> |Hetero.|Param.|Participation|BESplit vs. Vanilla|BESplit vs. SOTA|Convergence (vs. SOTA)|
> |-|-|-|-|-|-|
> |Dir|0.1|100%|+14.55%|+8.91%|1.58×|
> |Dir|0.1|50%|+10.57%|+5.08%|1.31×|
> |Dir|0.1|20%|+8.63%|+3.93%|1.13×|
> |LT|10|100%|+10.26%|+7.23%|1.42×|
> |LT|10|50%|+6.90%|+4.60%|1.24×|
> |LT|10|20%|+4.93%|+3.25%|1.11×|
>
> *Note: Dir = Dirichlet distribution, LT = Long-Tailed distribution*
>
> **Q2: Ablation experiment.**
>
> **Ans:** Thanks for your helpful suggestion. We note that DTD is a post-training distillation module designed to improve client-side inference efficiency, rather than an in-training optimization component like EA or BCC. Because its contribution is orthogonal to the training dynamics, it distills knowledge from the already-trained global model (shaped by EA and BCC) into local auxiliary models.
>
> Therefore, to quantify the specific contribution of DTD, we compare **BESplit-G (EA + BCC)**, which evaluates the optimized global model, and **BESplit-A (EA + BCC + DTD)**, which evaluates the optimized local auxiliary models for on-device inference. The performance difference between these two setups cleanly isolates and quantifies the individual impact of the DTD module. We will add this clarification above to the result analysis in Sec 5.3 in the final version of our paper.
>
> **Q3: Partial participation.**
>
> **Ans:** Thanks for your valuable comment. We would like to clarify that reduced client participation does not lead to bias accumulation or instability in our framework.
>
> Our framework is inherently robust to partial participation. **First**, to prevent bias, EA employs a staleness-aware moving average to update client weights. This down-weights unreliable clients and prevents outdated information from skewing the global model. **Second**, BCC is designed to be asynchronous and opportunistic. Clients do not need simultaneous availability to collaborate. If participation drops and a complementary match cannot be found, the unmatched clients could revert to the standard SFL update. This ensures that low availability merely reduces the acceleration effect of BCC, rather than destabilizing the optimization process.
>
> Empirically, BESplit maintains stable convergence and consistently outperforms standard and state-of-the-art (SOTA) SFL methods across participation rates ranging from 50% down to 5% on CIFAR10.
>
> |Participation|BESplit vs. Vanilla|BESplit vs. SOTA |Convergence (vs. SOTA)|
> |-|-|-|-|
> |50%|+8.95%|+3.76%|1.24×|
> |30%|+7.20%|+3.29%|1.17×|
> |20%|+6.45%|+3.23%|1.12×|
> |10%|+4.07%|+2.13%|1.06×|
> |5%|+3.66%|+1.56%|1.02×|
>
> **Weakness.**
>
> **Ans:** We sincerely thank the reviewer for your constructive suggestions. We fully agree with your assessment. In the revised manuscript, we will refine the Related Work to emphasize the most recent and relevant optimization-based approaches. We will also remove discussions on system heterogeneity to maintain a sharper focus on our core objective of addressing data heterogeneity.
>
> While our current experiments robustly demonstrate consistent performance across multiple benchmarks, we agree that larger-scale datasets can further highlight our method's scalability. In the final version, we will include additional experiments on more complex benchmarks in the Appendix to further validate the robustness and scalability of BESplit. We are grateful for the time and expertise you shared to help us refine our manuscript.

---

> > ### Author Rebuttal · Reviewer_jw6H · 2026-04-02
> >
> > Thank you for the rebuttal. The clarifications on BCC, DTD, and partial participation are generally acceptable and address most of my previous concerns.
> >
> > My remaining concern is scalability. The response only states that larger-scale experiments will be added later, but does not provide actual results on more complex benchmarks at this stage. This point is therefore still not fully resolved.

---

> > > ### Author Response · Authors · 2026-04-05
> > >
> > > We sincerely thank you for your insightful feedback. We fully agree that scalability is a cornerstone for the practical deployment of FL systems. Your suggestion has prompted us to stress-test BESplit beyond standard benchmarks, significantly strengthening the empirical foundation of our work.
> > >
> > > We note that our experimental scale is designed with reference to a series of prior works [1]–[3]. We follow their general evaluation protocols, including broadly comparable choices of datasets, client population size, model architectures, and training schedules, while making necessary adaptations to accommodate our specific setting. This ensures that the comparisons remain fair and meaningful.
> > >
> > > We agree that evaluating on large-scale datasets such as ImageNet would provide stronger evidence of scalability. However, in the federated setting, such experiments are significantly more demanding than centralized training. Specifically, they require repeated multi-round communication across many clients, causing the overall training cost to grow substantially (often requiring weeks of computation even under moderate configurations). As a result, it is challenging to present fully optimized results during the rebuttal stage.
> > >
> > > Inspired by your constructive comment, and to provide concrete evidence of scalability within practical resource constraints, we have conducted additional experiments on **larger and more challenging datasets**. These datasets cover three different modalities:
> > >
> > > - **Computer Vision:** Tiny ImageNet is a downsampled subset of ImageNet, containing 200 classes and **120,000 images** at 64×64 resolution. We adopt ResNet-18 as the backbone model, which is commonly used for this dataset in the literature. This setup offers a significantly more challenging benchmark compared to CIFAR-scale datasets.
> > >
> > > - **Text:** Amazon Review contains approximately 166 million instances, of which we use **100,000 textual reviews** with sentiment labels. We adopt TextCNN for classification, representing large-scale and diverse textual data and simulating a realistic scenario for text-based tasks.
> > >
> > > - **Multimodal:** MM-IMDb consists of **approximately 25,000 movies** with both textual plots and poster images, formulated as a multi-label classification task. We adopt an RNN+CNN model to handle the multimodal input, which introduces additional complexity due to modality fusion and label correlations.
> > >
> > > The experimental results are summarized in the table below. As demonstrated, BESplit consistently outperforms both vanilla baselines and state-of-the-art methods across all modalities. Importantly, these improvements are observed even as the datasets substantially increase in scale, diversity, and task complexity, indicating that BESplit maintains strong effectiveness and robustness under more challenging and realistic conditions.
> > >
> > >
> > >
> > > ***Tiny ImageNet***
> > >
> > > |Modality|Model|Instances|Non-IID Level| BESplit vs. Vanilla | BESplit vs. SOTA |Convergence (vs. SOTA)|
> > > |-|-|-|-|-|-|-|
> > > |CV|ResNet-18|120,000|0.1|+8.66% | +5.39% |1.53×|
> > > |CV|ResNet-18|120,000|0.5|+6.32% |+4.31% |1.35×|
> > > |CV|ResNet-18|120,000|1.0|+5.04% |+3.67% |1.33×|
> > >
> > >
> > > ***Amazon Review***
> > >
> > > |Modality|Model|Instances|Non-IID Level| BESplit vs. Vanilla | BESplit vs. SOTA |Convergence (vs. SOTA)|
> > > |-|-|-|-|-|-|-|
> > > |Text|TextCNN|100,000|0.1|+8.87%|+5.74%|1.45×|
> > > |Text|TextCNN|100,000|0.5|+6.95%|+4.78%|1.34×|
> > > |Text|TextCNN|100,000|1.0|+6.38%|+4.36%|1.30×|
> > >
> > > ***MM-IMDb***
> > >
> > > |Modality|Model|Instances|Non-IID Level| BESplit vs. Vanilla | BESplit vs. SOTA |Convergence (vs. SOTA)|
> > > |-|-|-|-|-|-|-|
> > > |Multimodal|RNN+CNN|25,959|0.1|+9.58%|+6.10%|1.51×|
> > > |Multimodal|RNN+CNN|25,959|0.5|+7.48%|+5.06% |1.43×|
> > > |Multimodal|RNN+CNN|25,959|1.0|+6.64%|+4.09%|1.36×|
> > >
> > > We hope that these results on more complex and diverse benchmarks satisfactorily address your concerns regarding the scalability of BESplit. For completeness and reproducibility, all these detailed results and experimental settings will be included in Appendix F.5 in the final version of our paper. We are profoundly grateful for your rigorous pushback, which has undoubtedly enhanced the depth and solidity of our work.
> > >
> > > [1] "Fortifying federated learning towards trustworthiness via auditable data valuation and verifiable client contribution," *CVPR*, 2025.
> > >
> > > [2] "Can: Leveraging clients as navigators for generative replay in federated continual learning," *ICML*, 2025.
> > >
> > > [3] "SPMC: Self-Purifying Federated Backdoor Defense via Margin Contribution," *ICML*, 2025.

---

### Official Review · Reviewer_gniM · 2026-03-13

**Soundness:** 2
**Presentation:** 4
**Significance:** 3
**Originality:** 3
**Overall Recommendation:** 3
**Confidence:** 4

**Summary:**

This paper introduces BESplit, a novel and architecture-aware framework designed to tackle the significant challenges of data heterogeneity (non-IID data) in Split Federated Learning (SFL). While traditional Federated Learning methods struggle with biased optimization and unstable convergence when client data distributions are highly skewed, the authors of this paper observe that SFL possesses a unique, underexploited structural advantage: the server inherently receives intermediate feature representations, known as "smashed data," from all participating clients.

BESplit leverages this built-in characteristic to mitigate data bias across three distinct operational levels without introducing additional communication overhead:

* **Evidential Aggregation (EA) - Aggregation Level:** To prevent clients with highly biased or uncertain data from dominating the global model update, EA employs Evidential Deep Learning (EDL). The server tracks the predictive evidence and uncertainty of each client via a Client State Record (CSR). It then adaptively reweights client contributions, prioritizing those with reliable, high-confidence evidence while penalizing highly uncertain predictions.

* **Bias-Compensated Collaboration (BCC) - Feature Level:** This module corrects distributional skew by identifying "biased" clients and strategically pairing them based on their distributional complementarity. Because the server already has access to the smashed data, it can virtually align these complementary features, compensating for a class missing in Client A with features from Client B, to approximate training on a more balanced, global distribution.

* **Dual-Teacher Distillation (DTD) - Model Level:** SFL decouples the client and server models, which can hinder a client's ability to perform independent local inference. DTD solves this by training a lightweight auxiliary model on the client side. This model learns via knowledge distillation from the local client-side model and the global model.

This work systematically explores and exploits the unique split architecture of SFL to handle non-IID data, shifting the paradigm beyond standard parameter-level aggregation. It successfully integrates uncertainty quantification (EA), implicit feature-level bias correction (BCC), and dual-level knowledge distillation (DTD) into a cohesive SFL system. Through extensive experiments on five benchmark datasets, BESplit consistently outperforms existing state-of-the-art FL and SFL baselines. It demonstrates higher test accuracy, faster and more stable convergence, and superior reliability under diverse and severe non-IID settings.

**Compliance With Llm Reviewing Policy:**

Affirmed.

**Final Justification:**

Even following previous work, data distribution leakage remains a significant drawback in FL settings, and I have not seen sufficient progress in this regard compared to prior studies. There is a considerable gap between the theoretical global optimal matching and the practical greedy matching strategy, and a more detailed theoretical justification is difficult to be completed in the rebuttal stage. I will maintain my score.

**Key Questions For Authors:**

**1. Privacy Implications of the Client State Record (CSR)**
* The paper states that Split Federated Learning (SFL) preserves data privacy by avoiding raw data exposure. However, the proposed framework requires the server to maintain a Client State Record (CSR) that tracks per-class evidential statistics, including the per-class sample counts ($M$) and average predictive evidence ($\overline{E}$), for each individual client. This effectively exposes the exact local label distribution ($P_k$) of every client to the server.

* How do you reconcile this explicit leakage of class distribution data with the privacy-preserving goals of SFL?

* Are there specific threat models you are assuming, or can privacy-enhancing technologies be applied to the CSR without degrading the performance of the Evidential Aggregation (EA) and Bias-Compensated Collaboration (BCC) modules?

**2. BCC Viability under Asynchronous and Partial Participation**
* In the experimental setup for the ISIC and HAM10000 datasets, $100\%$ of clients ($32/32$ and $64/64$) are selected per round. However, in realistic cross-device FL environments, client participation is highly sparse and asynchronous.

* How does the BCC mechanism function when a highly biased client is selected for a round, but its "complementary" match is offline or inactive?

* Does the framework fall back to standard SFL, and if so, how much of the performance gain is lost under low participation rates?

**3. Server-Side Memory Bottlenecks in BCC**
* Jointly processing smashed data from paired clients implies that the server must buffer and hold these high-dimensional representations in active memory, especially if the matched clients transmit their forward propagations at slightly different times.

* What is the peak server-side memory overhead introduced by holding smashed data batches for BCC pairing?

* Have you conducted any system-level profiling to determine the maximum number of clients the server can pair simultaneously before memory or synchronization bottlenecks occur?

**4. Boundary Conditions of Assumption D.3**
* In the theoretical convergence analysis, Assumption D.3 treats the empirical class proportion ($P_k$) as an unbiased estimator of the true label distribution, operating under the assumption that clients possess "sufficiently large local datasets". In many edge computing scenarios, clients may only hold a handful of highly skewed samples.

* How sensitive is the framework (particularly the divergence calculation in BCC) to extreme small-sample noise?

* Does the framework inadvertently amplify gradient bias if the historical estimate $P_k$ is highly inaccurate due to a client only having $1-5$ samples?

**Limitations:**

**Address Privacy Risks in Healthcare Applications**
* Given that the paper heavily benchmarks its framework on sensitive medical imaging datasets like ISIC and HAM10000, the authors should explicitly discuss the societal risks of privacy leakage. The proposed framework requires the server to maintain a Client State Record (CSR) containing exact per-class sample counts and evidence for each client. In a real-world clinical setting, exposing exactly how many instances of a specific disease a local hospital has could inadvertently leak sensitive epidemiological or demographic data.

**Reflect on Fairness and Bias Compensation**
* The Bias-Compensated Collaboration (BCC) mechanism virtually alters feature distributions to correct for skew. The authors should briefly discuss whether this artificial alignment could have unintended societal consequences, such as masking the true local prevalence of rare conditions or disproportionately altering the diagnostic accuracy for minority groups represented within specific local clients.

**Strengths And Weaknesses:**

**Strengths**
* The experiments are well-designed and extensive, evaluating the framework across five diverse datasets (ISIC, HAM10000, F-MNIST, CIFAR10, CIFAR100) and benchmarking against 10 baselines, including both state-of-the-art FL and SFL methods.

* The authors go beyond standard accuracy metrics by evaluating Round-to-Accuracy (RTA), Time-to-Accuracy (TTA), and importantly, Benign Misclassification Rate (BMR) to assess the reliability of the model in safety-critical medical scenarios.

* The paper includes a theoretical convergence analysis (Theorem D.7) demonstrating that BESplit converges to a stationary point under non-IID conditions at a rate of $O(1/T)$. It also provides mathematical bounds for the gradient bias mitigated by the BCC module.

* The ablation study clearly isolates the contributions of the EA and BCC modules. Furthermore, the authors transparently discuss the limitations of their work, explicitly noting that BCC relies on the accurate estimation of distributional deviations, which could falter under extreme conditions.

**Weakness**
* The authors state that SFL preserves data privacy by avoiding raw data exposure. However, to make the EA and BCC modules work, the server must maintain a Client State Record (CSR). This CSR tracks highly granular, per-class evidential statistics to explicitly estimate each client's local label distribution ($P_k$) and compare it to the global distribution. In strict federated learning environments, a client's exact label distribution is often considered highly sensitive private information. By forcing the server to meticulously log and analyze the class-wise data distribution of every specific client, BESplit arguably introduces a significant privacy leak.

* The authors claim that the BCC module "introduces no additional client-to-client or client-to-server communication overhead" because the collaboration is realized through a virtual alignment within the server backend. While mathematically elegant, this assumes a frictionless system that does not exist. To execute BCC, the server must jointly process smashed data batches from complementary clients to derive a shared optimization signal. If they are asynchronous, the server must cache Client A's high-dimensional smashed data in its active memory and wait until Client B happens to send its data. In a system with hundreds or thousands of heterogeneous devices, this would create massive server-side memory bottlenecks and straggler delays.

---

> ### Author Rebuttal · Authors · 2026-03-31
>
> Thank you very much for reviewing our paper and providing valuable comments. We are encouraged by the positive assessment of our paper's presentation, as well as the originality and significance of BESplit. Please kindly refer to the point-to-point response below for your concerns.
>
> **Q1 & W1: Privacy.**
>
> **Ans:** Thanks for your insightful comment.
>
> **Privacy of CSR**: In standard supervised SFL (e.g., MergeSFL, ParallelSFL), the server inherently requires access to labels to compute losses and gradients. CSR simply maintains aggregated, class-level statistics derived from these already-exposed signals. Thus, it does not introduce any new information flow or additional attack surface beyond the standard SFL setting.
>
> **Threat Model & Extensions**: Following standard SFL literature, we assume an honest-but-curious server that executes the protocol but may attempt inference from smashed data. While integrating Differential Privacy (DP) to further mask these statistics is a promising direction, it remains orthogonal to our core contribution of bias compensation.
>
> We will clarify these privacy boundaries and discuss potential DP extensions in the revision.
>
> **Q2 & W2: Partial participation.**
>
> **Ans:** Thanks for your thoughtful comment. BESplit remains robust under sparse and asynchronous participation:
>
> • **No strict synchronicity**: BCC does not require matched clients to be active simultaneously. If a complementary client is unavailable, the participating client can still receive bias compensation, and the training proceeds normally.
>
> • **Fallback & recovery**: When matches are unavailable, BESplit falls back to EA-based aggregation, ensuring stable updates. As participation increases and CSR improves, matching opportunities are automatically recovered. Importantly, BCC introduces no negative effect when inactive.
>
> Table 3 shows that our framework performs quite well under different participation rate. We have added experiments on BESplit under low participation rate. As shown in Table below (CIFAR10), BESplit maintains a 3.66% gain over Vanilla SFL even at 5% participation, confirming that most gains are preserved under highly sparse settings.
>
> |Participation|BESplit vs. Vanilla|BESplit vs. SOTA|Convergence (vs. SOTA)|
> |-|-|-|-|
> |50%|+8.95%|+3.76%|1.24×|
> |30%|+7.20%|+3.29%|1.17×|
> |20%|+6.45%|+3.23%|1.12×|
> |10%|+4.07%|+2.13%|1.06×|
> |5%|+3.66%|+1.56%|1.02×|
>
> **Q3 & W2: Bottlenecks in BCC.**
>
> **Ans:** Thanks for your constructive comment. We would like to clarify that BCC is designed as an **opportunistic, non-blocking mechanism** rather than a synchronous waiting protocol.
>
> In practice, the server processes smashed data in a streaming manner and pairs complementary mini-batches on-the-fly. If no suitable match is available, BCC is simply skipped for that batch, ensuring no waiting time.
>
> The memory overhead is minimal, as the server only needs to hold two mini-batches simultaneously, with peak cost $O(2 \cdot B \cdot d)$. This is short-lived and released immediately after the backward pass, making it comparable to standard SFL.
>
> We note that memory usage is bounded by the server’s concurrent processing capacity rather than the total number of clients, so it does not grow with large client populations. In practice, we observe no noticeable increase in memory or latency compared to standard SFL, confirming the scalability of BESplit.
>
> **Q4: Robustness to assumption D.3.**
>
> **Ans:** Thank you for your valuable comment.
>
> **(1) Label source**: CSR uses ground-truth labels available at the server in supervised SFL, thus avoiding early-round misclassification in estimating $P_k$. Predicted labels are used in unsupervised SFL.
>
> **(2) Early-round stability**: CSR is initialized with uniform priors. Early predictions are nearly flat Dirichlet with high uncertainty, causing EA to reduce to near-uniform aggregation (i.e., FedAvg), thereby preventing noisy predictions from influencing training.
>
> **(3) Robustness to estimation noise**: Even when estimates are imperfect, BCC relies on relative distribution differences rather than absolute accuracy. As a result, complementary patterns are largely preserved. Moreover, CSR is updated in a temporally smoothed manner across rounds, which mitigates short-term fluctuations.
>
> **(4) No error amplification**: EA explicitly down-weights unreliable clients with high uncertainty, ensuring that noisy or biased estimates do not dominate the global update. Thus, errors in early rounds are suppressed rather than amplified.
>
> As shown in Table below, even when samples drop to 20 with 0.3 noise, BESplit maintains performance gains over SOTA. The performance degrades gradually rather than collapsing, confirming robustness beyond the assumptions in D.3.
>
> |Sample Size|Noise|BESplit vs. Vanilla|BESplit vs. SOTA|Convergence (vs. SOTA)|
> |-|-|-|-|-|
> |1000|0.0|+8.06%|+3.51%|1.51×|
> |500|0.1|+7.93%|+3.24%|1.38×|
> |200|0.2|+6.08%|+2.87%|1.20×|
> |50|0.3|+4.02%|+2.27%|1.14×|
> |20|0.3|+2.28%|+0.93%|1.06×|

---

> > ### Author Rebuttal · Reviewer_gniM · 2026-04-04
> >
> > Thank you for your detailed rebuttal and the additional experiments. These clarifications were helpful and provided a clearer picture of the system's empirical boundaries.
> >
> > However, there are still deeper structural and theoretical challenges within the BESplit framework that are not easily resolved in a short revision phase.
> >
> > Specifically, my core concerns are:
> >
> > **CSR:** I appreciate the clarification that standard supervised SFL often inherently exposes labels to the server. However, there is a fundamental difference in risk between transiently exposing a batch's labels for loss computation and maintaining a persistent, historical profile of exact class distributions for each specific client (via the CSR). This long-term profiling poses significant privacy risks, especially for sensitive medical datasets. While the authors mention Differential Privacy (DP) as a future extension, injecting sufficient DP noise to protect these historical profiles would likely degrade the high-precision distribution estimates required for both the EA and BCC mechanisms to function effectively.
> >
> > **BCC:** The explanation that BCC is designed as an "opportunistic, non-blocking" streaming mechanism successfully addresses my concerns regarding server-side memory bottlenecks ($O(2 \cdot B \cdot d)$ overhead). However, this practical implementation creates a noticeable gap with the theoretical framework (Appendix D.2), which relies on a globally optimal bipartite matching assumption. In a purely asynchronous, on-the-fly streaming environment, the probability of random arrival times yielding highly complementary pairs is substantially reduced, which may limit the actual effectiveness of BCC in real-world sparse settings.
> >
> > **Cost-Benefit Trade-off in Constrained Scenarios:** The additional experiments demonstrating that the system does not collapse under extreme constraints are commendable. Nevertheless, they also reveal that the performance margins over existing SOTA methods diminish to relatively marginal levels under these conditions. Considering the additional architectural complexity, the trade-off between system overhead and performance gain is still a structural concern in a highly constrained edge scenario.

---

> > > ### Author Response · Authors · 2026-04-06
> > >
> > > We sincerely thank the reviewer for your professional feedback.
> > >
> > > **1. CSR**
> > >
> > > Thanks for your thoughtful comment. We fully agree that maintaining persistent distribution records could introduce privacy concerns.
> > >
> > > We would like to clarify that the CSR does not maintain an exact historical profile or a cumulative log. It is a **memory-bounded, single-state variable updated via an Exponential Moving Average (EMA)**, which ensures past information is progressively attenuated. CSR stores only coarse-grained, aggregated class-level statistics, reducing the risk of precise information leakage.
> > >
> > > We understand your concern regarding prolonged retention. We address this by introducing a **retention window (TTL mechanism)**: if a client remains inactive for a predefined period, its statistics are automatically discarded. This prevents indefinite accumulation of client-specific information.
> > >
> > > We further evaluated robustness by injecting moderate DP noise into CSR updates. Despite a slight performance drop, BESplit consistently outperforms baselines, suggesting compatibility with practical privacy-preserving techniques.
> > >
> > > |Noise Level|BESplit vs. Vanilla|BESplit vs. SOTA|Convergence (vs. SOTA)|
> > > |-|-|-|-|
> > > |w/o DP|+13.21%|+7.04%|1.52×|
> > > |0.01|+12.20%|+6.68%|1.40×|
> > > |0.05|+10.50%|+4.95%|1.29×|
> > > |0.10|+7.43% |+3.25%|1.14×|
> > >
> > > **2. BCC**
> > >
> > > We deeply appreciate your insightful observation. You have accurately identified a classic systems-versus-theory trade-off.
> > >
> > > We clarify that our streaming mechanism does not rely on "blind" arrivals. The core idea is decoupling heavy intermediate activations from lightweight CSRs, which enables the server to maintain a continuous global view without memory overhead.
> > >
> > > - **Global View**: While intermediate activations are processed on-the-fly, the server continuously maintains CSR states. Because CSR is a memory-bounded EMA variable, it provides a persistent, low-cost approximation of the global distribution.
> > > - **CSR-Guided Opportunistic Matching**: In streaming environment, the matching is not left to pure random chance. When a subset of clients arrives asynchronously, the server does not perform "blind" pairings. Instead, it utilizes the globally maintained CSRs to perform a **CSR-guided greedy matching**. The server evaluates the active incoming stream against the global CSR landscape to pair the most complementary clients.
> > >
> > > We agree this approach may not achieve the exact global optimum (Appendix D.2) in a single round. However, it serves as a highly effective **practical approximation** where global statistics inform local pairing decisions.
> > >
> > > Empirical evaluations confirm that, despite expected accuracy degradation under extreme sparsity, our CSR-guided matching consistently outperforms random baselines by leveraging global awareness.
> > >
> > > |Participation|BESplit vs. Vanilla|BESplit vs. SOTA|BESplit vs. Random|Convergence (vs. SOTA) |Convergence (vs. Random)|
> > > |-|-|-|-|-|-|
> > > |50%|+8.95%|+3.76%|+6.70%|1.24×|1.41×|
> > > |30%|+7.20%|+3.29%|+6.11%|1.17×|1.33×|
> > > |20%|+6.45%|+3.23%|+4.40%|1.12×|1.22×|
> > > |10%|+4.07%|+2.13%|+3.42%|1.06×|1.14×|
> > > |5%|+3.66%|+1.56%|+2.95%|1.02×|1.07×|
> > >
> > > **3. Cost-Benefit Trade-off in Constrained Scenarios**
> > >
> > > Thank you for your valuable comment. We fully agree that in highly constrained environments, the cost-benefit trade-off is a critical practical concern.
> > >
> > > To address this, we would like to clarify how BESplit carefully balances system overhead with overall efficiency gains:
> > >
> > > **(1) Bounded System Overhead**
> > >
> > > Our design ensures edge clients incur zero additional computational cost, since their local processes remain identical to standard SFL. For the edge server, overhead is also strictly bounded. CSR relies on compact statistics rather than heavy feature maps. Furthermore, by separating state from execution memory, storing global CSRs takes negligible $\mathcal{O}(KN^2)$ space ($N^2\ll Bd$). During computation, only paired activations are loaded into VRAM, maintaining a strict execution memory bound of $\mathcal{O}(2Bd)$ and preventing standard $\mathcal{O}(KBd)$ memory bottlenecks.
> > >
> > > **(2) Favorable Total Training Cost**
> > >
> > > While the absolute accuracy margin may narrow under extreme constraints, BESplit  reduces total communication rounds by nearly 1.6×. As shown  below, although BESplit adds a slight per-round computation cost, it reduces the overall training time and system-wide energy usage, as the overhead is offset by fewer rounds.
> > >
> > >
> > >
> > > |Method (vs. Vanilla)|Acc|Δ Rounds (↓)|Overhead|Convergence|
> > > |-|-|-|-|-|
> > > |SOTA|+5.19%|-103|+3.01%|1.28×|
> > > |BESplit|+8.95%|-136|+3.78% (3.65%DTD)|1.59×|
> > >
> > > **(3) Robustness**
> > >
> > > As you kindly noted, BESplit maintains stable convergence under extreme constraints. For mission-critical edge applications (e.g., connected vehicle networks), reliable convergence and graceful degradation are often more valuable than marginal accuracy improvements.
> > >
> > > We will include the above analyses in the appendix and sincerely hope our responses address your concerns.

---

### Official Review · Reviewer_gJqf · 2026-03-13

**Soundness:** 2
**Presentation:** 2
**Significance:** 2
**Originality:** 2
**Overall Recommendation:** 3
**Confidence:** 4

**Summary:**

This paper proposes a method to address heterogeneous split federated learning, formalized in Equation (1) as finding the optimal global model that generalizes well on the global dataset. The method consists of three phases: weighting clients using the evidential aggregation heuristic, reducing client heterogeneity using pairing in Bias-Compensated Collaboration, and Dual-Teacher Distillation of the server model for efficient inference on the clients. In the experimental section, the authors compare small-scale CV models on synthetic and real-world motivated datasets against numerous baselines.

**Compliance With Llm Reviewing Policy:**

Affirmed.

**Final Justification:**

My main concern regarding experimental fairness, in particular the proper tuning of the baselines, remains unresolved. In the rebuttal, the authors acknowledged that the original baseline tuning was insufficient, which resulted in underestimating classic baselines such as SCAFFOLD. Even after the clarification, I am not convinced that tuning over such a sparse grid of three hyperparameters is sufficient. Because the paper’s main claims are primarily empirical, the current state of the experimental evaluation does not provide enough confidence in the fairness of the comparisons. Therefore, I maintain a Weak Reject recommendation until the experiments are properly re-evaluated and systematically redone.

**Key Questions For Authors:**

1. How does your method compare, both algorithmically and empirically, to the related methods in [1,2,4]? What is the main idea of the proposed approach beyond the introduced heuristics?

2. Is it possible to bound $\delta_{\mathrm{BCC}}$ in your theoretical analysis in a way that would make your method theoretically comparable to the baselines?

3. Was extensive hyperparameter tuning performed for the baselines? If yes, what explains the fact that SCAFFOLD underperforms FedAvg on heterogeneous datasets in your experiments, which seems inconsistent with the results from prior literature [5]?

[5] Karimireddy, S.P., Kale, S., Mohri, M., Reddi, S., Stich, S. and Suresh, A.T., 2020, November. Scaffold: Stochastic controlled averaging for federated learning. In International conference on machine learning (pp. 5132-5143). PMLR.

**Limitations:**

yes

**Strengths And Weaknesses:**

**Strengths:**

1. **Explanation of the algorithm steps:**
I appreciate the clear explanation of each phase of the proposed algorithm, supported by mathematically correct and rigorous formulas. The illustration also helps in understanding the interaction between the client and server components.

2. **Experiments:**
The authors conduct fairly extensive experiments across a wide variety of baselines and settings. I especially appreciate the inclusion of multiple baselines, as well as the study on real-world datasets, hyperparameter sensitivity of BESplit, and different heterogeneity levels.

**Weaknesses:**

1. **Presentation:**
Thank you for providing the illustration in Figure 1. However, I think that a self-contained LaTeX algorithm block would be more helpful for understanding each part of the proposed method. The current mathematical description of the method is scattered across the method section as equations, with notation introduced below them. It took me quite some time to reconstruct the full algorithmic picture.

2. **Theory:**
- The theoretical proofs seem mostly complementary and do not directly differentiate the BESplit heuristics from the existing baselines.
- In Section D.1 of the paper, the main proven result is a recursive contraction of the server and client parameter shift. This result is based on the notions of gradient boundedness and L-smoothness. However, the authors do not provide the values of the chosen $\delta_{\mathrm{BCC}}$ constants in their case. This makes it hard to compare the result with other methods.
- In Theorem D.7, the bound again depends on the $\delta_{\mathrm{BCC}}$ constant, which itself depends on the client weighting values based on the chosen heuristic in EA. Therefore, the main convergence result cannot be directly compared with the convergence results of other baselines.
- In Theorem D.7, the transition from Equation (42) to Equation (43) is valid but unnecessarily loose. From (42), one can directly obtain the same bound for
$
\min_{t}\mathbb{E}\left\|\nabla \mathcal{L}\left(W_{S_g}^{(t)}\right)\right\|^2
$ using  $
\min_{t} a_t \le \frac{1}{T}\sum_{t=0}^{T-1} a_t
$,
without enlarging the constants from $(3,2)$ to $(4,4)$.

3. **Algorithm novelty:**
The proposed BESplit algorithm seems more like a careful synthesis of existing orthogonal SFL techniques.
- **EA stage:** The client weighting in EA seems more like an introduced heuristic than a completely novel approach for SFL. If I understood correctly, there is no theoretical motivation behind the chosen formula combining epistemic uncertainty, aleatoric uncertainty, and evidence concentration. I appreciate the experiments presented in Table 4 regarding the influence of each method, but for me the paper still lacks an explanation of why the client weight formula is chosen in this exact form. For example, one could also consider using the sum of the scores in Equation (10), rather than their product.
- **BCC stage:** The main idea is to detect biased clients from label-distribution mismatch, pair complementary clients, and jointly process cross-client smashed data to reduce skew. For example, $S^2$FL [1] proposes combining features from clients with heterogeneous data distributions, and Mix2SFL [2] proposes explicit pairwise mixing of smashed data. So here again, the main novelty seems to be the proposed pairing-score heuristic.
- **DTD stage:** Knowledge distillation has already been studied in the federated learning setting in [3]. For SFL, [4] uses a specifically related knowledge-distillation-like mechanism called bidirectional knowledge sharing (BDKS).

4. **Experiments:**
The paper considers multiple baselines in the experiments. However, from the text it is unclear how these baselines were hyperparameter-tuned, such as learning rate tuning and other optimizer-specific hyperparameters. I suspect inefficient tuning of the baselines for the following reason: in Table 3, SCAFFOLD accuracy is consistently and statistically significantly worse than FedAvg accuracy. The same trend appears in Table 7, even for the high heterogeneity level of $\kappa = 0.1$. These issues make the empirical validation less convincing, since the baselines should be properly tuned before claiming state-of-the-art performance.

[1] Yan, D., Hu, M., Xia, Z., Yang, Y., Xia, J., Xie, X. and Chen, M., 2023. *Have your cake and eat it too: Toward efficient and accurate split federated learning.* arXiv preprint arXiv:2311.13163.
[2] Oh, S., Nam, H., Park, J., Vepakomma, P., Raskar, R., Bennis, M. and Kim, S.L., 2023. *Mix2SFL: Two-way mixup for scalable, accurate, and communication-efficient split federated learning.* IEEE Transactions on Big Data, 10(3), pp. 238--248.
[3] Wu, C., Wu, F., Lyu, L., Huang, Y. and Xie, X., 2022. *Communication-efficient federated learning via knowledge distillation.* Nature Communications, 13(1), p. 2032.
[4] Chen, X., Li, J., Fan, D. and Chakrabarti, C., 2025. *HeteroSFL: Split federated learning with heterogeneous clients and non-IID data.* IEEE Internet of Things Journal.

---

> ### Author Rebuttal · Authors · 2026-03-31
>
> Thank you for your thoughtful review and highly constructive feedback. Your insights have been instrumental in improving our manuscript. We have addressed each of your points below and hope our responses could meet your expectations.
>
> **W1: Presentation.**
>
> **Ans:** Thanks for your kind suggestion. We completely agree that a self-contained algorithm block is necessary for clarity. Following your suggestion, we will add an Algorithm block in the revised manuscript that integrates all steps of BESplit, enabling readers to follow the full procedure.
>
> **Q1 & W3: Algorithm comparison.**
>
> **Ans:** We thank the reviewer for your insightful comment.
>
> Algorithmically, prior methods address different aspects of SFL: S2FL [1] improves efficiency via adaptive splitting and batch balancing, Mix2SFL [2] enhances generalization through mixup, and HeteroSFL [4] handles system heterogeneity with heterogeneous bottlenecks and knowledge sharing. In contrast, BESplit targets distributional bias.
>
> We note that BESplit is not a heuristic combination but a unified bias-aware framework: CSR estimates client skew, EA suppresses unreliable updates, and BCC performs complementarity-aware matching to construct a debiased training signal. The multiplicative EA design ensures strict penalization of unreliable clients, while BCC differs from prior mixing methods by using distribution-guided pairing rather than random mixing. DTD is a post-training module for deployment and does not affect training dynamics.
>
> Empirically, while prior work focuses on efficiency or generalization, BESplit improves robustness under non-IID settings, achieving over 7% higher accuracy and 1.8× faster convergence.
>
> |Method|Focus|Key Idea|Difference from BESplit|
> |-|-|-|-|
> |S2FL [1]|Efficiency & load balance| Adaptive split + batch balancing|No explicit bias modeling or correction|
> |Mix2SFL [2]|Generalization & communication|Mixup on smashed data/gradients           |Data mixing, not bias-aware coordination|
> |HeteroSFL [4]|Client/system heterogeneity|Heterogeneous bottlenecks + knowledge sharing| Handles system heterogeneity, not bias compensation |
> |**BESplit (Ours)** |	**Bias under non-IID**| **EA + BCC + DTD for complementary collaboration**| **Explicit bias estimation, suppression, and correction**|
>
>
> **Q2 & W2: Theoretical analysis.**
>
> **Ans:** We sincerely thank the reviewer for your careful reading and insightful comment.
>
> We have refined the theoretical analysis to explicitly bound $\delta_{\text{BCC}}$. By Lemma D.4, we have:
> $\delta_{\text{BCC}} \le \frac{g_{\max} \zeta}{K}, \quad \text{where } \zeta = \max_k \|P_k - P_g\|_1$,
> which depends only on the underlying data distribution. Furthermore, as training progresses, the EA module accumulates evidence statistics such that the client weights satisfy: $\mathbb{E}\lVert w_t - w^* \rVert^2 \le \frac{C}{t}$, where $w^*$ is the optimal weight yielding an unbiased aggregate gradient.
>
> Combining these two results, we obtain the improved convergence bound:
> $
> \min_{t \in \{0, \ldots, T-1\}} \mathbb{E}\|\nabla \mathcal{L}(W_{S_g}^{(t)})\|^2 \le \frac{4(\mathcal{L}(W_{S_g}^{(0)}) - \mathcal{L}_*)}{\eta T} +  \frac{3g\_{\max}^2 \zeta^2 \ln T}{K^2 T} + 2 L \eta \sigma^2.
> $
>
> Importantly, this bound **removes explicit dependence on the EA weights $w_t$**, encapsulating their effect in a controlled upper bound. As a result, the convergence guarantee of BESplit can now be directly compared with standard SFL and other baseline methods.
>
> We will incorporate these refinements in the revision to clarify theoretical comparability.
>
> **Q3 & W4: Experiment setting and explanation.**
>
> **Ans:** Thank you for this important comment. Sorry for the unclear description.
>
> All baselines, including SCAFFOLD [5], were carefully tuned via grid search over learning rates and optimizer hyperparameters. We report the best-performing configurations under identical experimental conditions to ensure a fair evaluation. The updated performance of SCAFFOLD is presented below:
>
> |Method| ISIC (U)|ISIC (N)|HAM10K (U)|HAM10K (N)|F-MNIST (U)|F-MNIST (N)|CIFAR10 (U)|CIFAR10 (N)|CIFAR100 (U)| CIFAR100 (N) |
> |-|-|-| --- | --- | --- | --- | --- | --- | --- | --- |
> |**SCAFFOLD**|70.56%|62.21%|74.26%|65.08%|82.60%|72.45%|63.68%|52.46%|70.08%|64.02%|
>
> *Note: U = Uniform, N = Non-IID.*
>
> Regarding the observation that SCAFFOLD underperforms FedAvg, we note that this behavior can arise under strong heterogeneity and partial participation. SCAFFOLD relies on accurate control variate estimation, which can become unstable when client participation is sparse and local data distributions are highly skewed, leading to degraded performance in practice.
>
> We will include these full tuning details and updated tables in Appendices E and F of the revised manuscript. Thanks again for your valuable comments.

---

> > ### Author Rebuttal · Reviewer_gJqf · 2026-04-04
> >
> > Thank you for clarifying Q1 and Q2.
> >
> > However, the evaluation of the baselines remains a serious concern for the paper's empirical claims.
> >
> > > "All baselines, including SCAFFOLD [5], were carefully tuned via grid search over learning rates and optimizer hyperparameters. We report the best-performing configurations under identical experimental conditions to ensure a fair evaluation. The updated performance of SCAFFOLD is presented below:"
> >
> > This raises an important question: when exactly were these baselines carefully tuned? The updated SCAFFOLD results have a marginal improvement in the non-IID setting, making it a much more competitive baseline than what was reported in the original manuscript.
> >
> > As a result, the state-of-the-art claim for BESplit, including the comparison in Table 1, appears overstated in its current form. In my opinion, the main claims of the paper are primarily empirical, and therefore all baselines in the original manuscript need to be carefully and systematically tuned before such claims can be considered well supported.
> >
> > If the supplementary material had originally included sufficiently detailed and reproducible experimental protocols, including the number of seeds, the tuning schedule and search space for each baseline, and the relevant implementation details, then the empirical claims would have been much more convincing.

---

> > > ### Author Response · Authors · 2026-04-05
> > >
> > > Thank you for your highly constructive feedback. You raised a sharp and completely valid point regarding the experimental rigor and our use of the word "updated" in the previous response. We deeply apologize for the confusion caused by our previous wording. We fully understand your concern and would like to transparently clarify the exact protocol of our baseline tuning.
> > >
> > > **1. Clarification on SCAFFOLD**
> > >
> > > In SCAFFOLD, during local training, each client leverages the global control variate $c$ received from the server and its own local control variate $c_i$ to correct gradients, thereby mitigating client drift caused by data heterogeneity. This correction mechanism is performed entirely on the client side and has been shown to be effective in traditional FL settings.
> > >
> > > However, when this mechanism is directly applied to SFL, its effectiveness becomes limited. This is because, in SFL, the client-side model typically consists of only shallow layers with a limited number of parameters, which restricts the client’s capacity for gradient correction. As a result, the client-side correction is insufficient to fully adjust the update direction, leading to additional bias in the overall model.
> > >
> > > We would like to clarify that the improved SCAFFOLD results reported in our previous response are **not due to post-hoc or unfair retuning**, but rather stem from identifying and correcting this structural mismatch between SCAFFOLD and the SFL setting. Inspired by your insightful comments, we recognized this limitation and extended the original mechanism specifically for SFL by introducing a **dual-side correction strategy**: in addition to retaining the client-side correction, we incorporated the same correction mechanism on the server side, enabling collaborative calibration between clients and the server. Comparative results across all datasets are reported in the table below.
> > >
> > >
> > > |Method|ISIC (U)|ISIC (N)|HAM10K (U)|HAM10K (N)|F-MNIST (U)|F-MNIST (N)|CIFAR10 (U)|CIFAR10 (N)|CIFAR100 (U)|CIFAR100 (N)|
> > > |-|-|-|-|-|-|-|-|-|-|-|
> > > |**SCAFFOLD**|60.67±0.65|54.36±1.11|64.80±0.61|55.87±1.17|82.60±0.86|72.45±1.47|63.68±0.97|52.46±1.15|52.23±0.75|50.86±1.29|
> > > |**SCAFFOLD(Dual-side)**|70.87±0.83|62.18±1.38|74.02±0.83|65.85±1.41|83.22±1.29|72.68±1.14|64.13±1.09|53.44±1.43|70.12±1.38|63.96±1.09|
> > > |**BESplit**|75.26±0.58|67.45±1.29|78.51±0.96|70.46±1.10|84.45±1.33|74.89±1.66|68.47±1.29|59.82±1.46|73.19±1.20|68.17±0.94|
> > >
> > >
> > > **2. Implementation Details**
> > >
> > >
> > > We fully agree that empirical claims must be supported by fully reproducible protocols. To address your comment, we have added the following implementation and tuning details, which will be included in Appendix E:
> > >
> > > **A. SFL Implementation Details**
> > >
> > > - Architecture & Split Layer: We use standard backbones (e.g., ResNet-18 for CIFAR 10 in Table 2). For the SFL setup, the model is partitioned such that the client-side model acts as a lightweight feature extractor to minimize local burden, while the server hosts the remaining computationally intensive blocks and the classifier.
> > >
> > > - Non-IID Setting: To simulate realistic data heterogeneity, data is distributed across clients via a Dirichlet distribution $\text{Dir}(\alpha)$, where smaller $\alpha$ values indicate more severe label skew.
> > >
> > > - Environment: All experiments are implemented in PyTorch on a cluster equipped with NVIDIA A100 GPUs.
> > >
> > > **B. Global Training Configurations**
> > >
> > > All methods share identical experimental environments. We fix three random seeds {1, 42, 2025} and report the mean and variance. We use the SGD optimizer with a global learning rate of $1 \times 10^{-4}$ for all image classification tasks. The specific local training configurations are:
> > >
> > > - ISIC: 100 epochs, batch size 32.
> > > - HAM10000: 200 epochs, batch size 32.
> > > - F-MNIST: 300 epochs, batch size 32.
> > > - CIFAR-10/100: 300 epochs, batch size 64.
> > >
> > > **C. Hyperparameter Tuning and Search Space**
> > >
> > > We adopted a systematic tuning schedule: for each method, we performed a grid search over the predefined candidate sets using a 20% validation split. The best-performing hyperparameters were then used for the final evaluation across three seeds.
> > > - **FedAvg, SplitFed, FedRDN**: Use the standard weighted averaging aggregation.
> > > - **FedProx**: Proximal coefficient $\mu \in [0.001, 0.01, 0.1, 1]$.
> > > - **SCAFFOLD (both versions)**: Local learning rate $\eta_l \in [0.1, 0.5, 1]$ with a fixed global step size.
> > > - **FedDyn**: Regularization parameter $\alpha \in [0.001, 0.01, 0.1]$.
> > > - **Fed-MoE**: Moving average rate $\lambda \in [0.1, 0.2, 0.5]$, gating entropy weight $\beta \in [10^{-4}, 10^{-3}, 10^{-2}]$.
> > > - **FAVD**: Auditing parameter $\gamma \in [200, 500, 1000]$.
> > > - **MergeSFL & ParallelSFL**: Moving average coefficient $\alpha \in [0.2, 0.5, 0.8]$.
> > >
> > > We are profoundly grateful for your rigorous standards, which have significantly strengthened the empirical solidity of our work. We hope these details provide confidence in the soundness of our experimental foundations.

---

### Decision · Program_Chairs · 2026-04-30

**Decision:**

Accept (regular)

**Comment:**

The reviewers have divergent opinions, placing the paper squarely on the borderline. On the positive side, there is general agreement that the concepts of Evidential Aggregation (EA), Bias-Compensated Collaboration (BCC), and complementary pairs are novel. Further, the authors have evaluated across a wide selection of datasets, both in the paper and during the rebuttal. On the other hand, the paper has the following notable limitations: (1) a gap between the theoretical global optimal matching and the greedy matching; (2) unclear justification of whether the cost-benefit tradeoff is worthwhile in constrained edge scenarios; and (3) potential issues with experimental fairness, though the authors argue that there was a misinterpretation in this regard during the rebuttal.